# Dualistic insulator states in 1T-TaS$_2$ crystals

Yihao Wang [1,13], Zhihao Li [1,2,13], Xuan Luo [3], Jingjing Gao [3], Yuyan Han [1], Jialiang Jiang[1], Jin Tang [4], Huanxin Ju [5], Tongrui Li [6], Run Lv [3,7], Shengtao Cui [6], Yingguo Yang [8], Yuping Sun [1,3,9], Junfa Zhu [6], Xingyu Gao [10], Wenjian Lu [3] ✉, Zhe Sun [6,9,11] ✉, Hai Xu [2,12] ✉, Yimin Xiong [4,11] ✉ & Liang Cao [1] ✉

While the monolayer sheet is well-established as a Mott-insulator with a finite energy gap, the insulating nature of bulk 1T-TaS$_2$ crystals remains ambiguous due to their varying dimensionalities and alterable interlayer coupling. In this study, we present a unique approach to unlock the intertwined two-dimensional Mott-insulator and three-dimensional band-insulator states in bulk 1T-TaS$_2$ crystals by structuring a laddering stack along the out-of-plane direction. Through modulating the interlayer coupling, the insulating nature can be switched between band-insulator and Mott-insulator mechanisms. Our findings demonstrate the duality of insulating nature in 1T-TaS$_2$ crystals. By manipulating the translational degree of freedom in layered crystals, our discovery presents a promising strategy for exploring fascinating physics, independent of their dimensionality, thereby offering a "three-dimensional" control for the era of slidetronics.

The discovery of Mott-insulating states holds significant relevance for unconventional superconductors and quantum spin liquid phase (QSL), showing promise for applications in quantum computing[1–4]. The Mott-insulating state in the layered octahedral 1T-TaS$_2$ crystals, first proposed four decades ago[5], has led to extensive investigations into the intertwined correlations among Mott-insulator, charge density wave (CDW), and superconducting states[6–9]. The Mott-insulating nature is widely accepted in isolated monolayer 1T-TaS$_2$ due to the preserved in-plane David-stars in the commensurate CDW (C-CDW) state and the absence of interlayer coupling[10–13]. However, in the past decade, a plot twist has emerged in understanding the insulating behavior in bulk 1T-TaS$_2$ crystals. Theoretical and experimental studies have

introduced the concept of the band-insulator mechanism, considering the key role of the David-stars dimerization associated with specific on-top David-stars T$_A$ stacking configurations (Fig. 1d)[10,13–18]. However, the theoretically predicted out-of-plane metallic state has not been observed in transport experiment[10,14–16,19]. And discrepancies exist between microscopic and macroscopic properties[17,20–22]. For instance, the recent findings of the band-to-Mott insulating phase transition at a narrow temperature window in 1T-TaS$_2$ crystals upon heating have not fully manifested in the macroscopic transport measurements yet[22,23]. Along with anomalies like the absence of long-range magnetic order and unexpected emergence of superconducting states[6,7,18,24–27], these complexities complicate the understanding of the physics of this

[1]Anhui Key Laboratory of Low-Energy Quantum Materials and Devices, High Magnetic Field Laboratory, HFIPS, Chinese Academy of Sciences, Hefei 230031, P. R. China. [2]Changchun Institute of Optics, Fine Mechanics and Physics, Chinese Academy of Sciences, Changchun, Jilin 130033, P. R. China. [3]Key Laboratory of Materials Physics, Institute of Solid State Physics, HFIPS, Chinese Academy of Sciences, Hefei 230031, P. R. China. [4]Department of Physics, School of Physics and Optoelectronics Engineering, Anhui University, Hefei 230601, P. R. China. [5]PHI Analytical Laboratory, ULVAC-PHI Instruments Co., Ltd., Nanjing 211110 Jiangsu, P. R. China. [6]National Synchrotron Radiation Laboratory, University of Science and Technology of China, Hefei 230026, P. R. China. [7]Science Island Branch of Graduate School, University of Science and Technology of China, Hefei 230026, P. R. China. [8]State Key Laboratory of Photovoltaic Science and Technology, School of Microelectronics, Fudan University, Shanghai 200433, P. R. China. [9]Collaborative Innovation Center of Advanced Microstructures, Nanjing University, Nanjing 210093, P. R. China. [10]Shanghai Synchrotron Radiation Facility (SSRF), Zhangjiang Laboratory, Shanghai Advanced Research Institute, Chinese Academy of Sciences, 239 Zhangheng Road, Shanghai 201204, P. R. China. [11]Hefei National Laboratory, Hefei 230028, P. R. China. [12]Center of Materials Science and Optoelectronics Engineering, University of Chinese Academy of Sciences, Beijing 100049, P. R. China. [13]These authors contributed equally: Yihao Wang, Zhihao Li. ✉e-mail: wjlu@issp.ac.cn; zsun@ustc.edu.cn; xuhai@ciomp.ac.cn; yxiong@ahu.edu.cn; lcao@hmfl.ac.cn

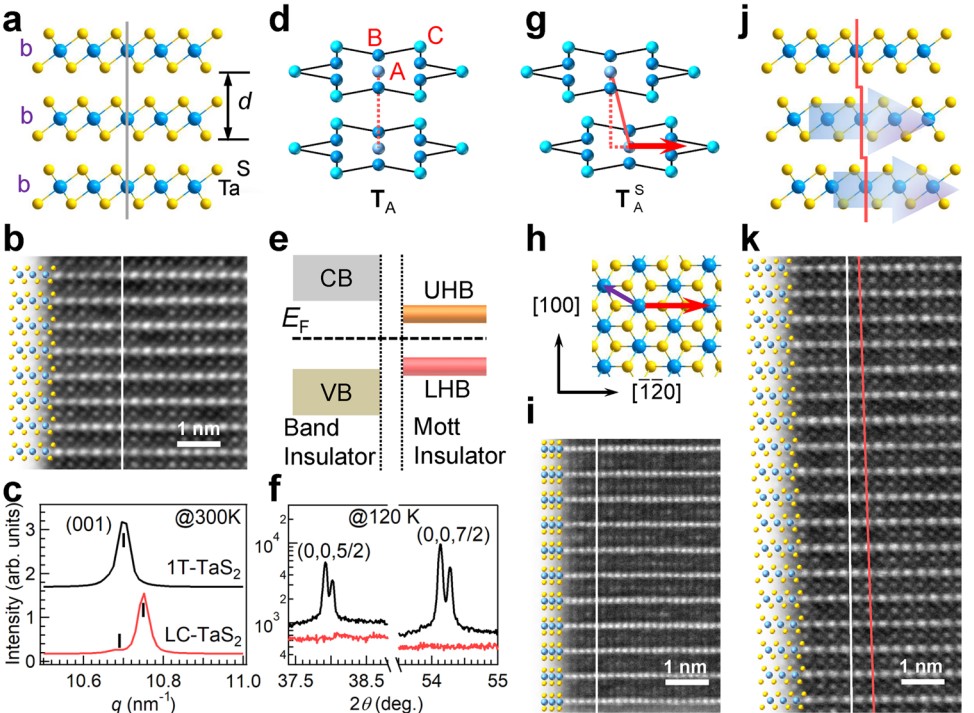

**Fig. 1 | Variation in the atomic structure and insulating nature. a** Schematic side view of 1T-TaS$_2$ crystals and (**b**) corresponding [100] cross-sectional HAADF-STEM image. **c** Synchrotron-based XRD spectra around the (001) diffraction peak collected at room temperature. **f** The out-of-plane superlattice and reflections for a 1T-TaS$_2$ crystal collected at 120 K. The peak splitting is attributed to the Cu K$_{\alpha 1}$ and K$_{\alpha 2}$ X-ray split. Illustration of (**d**) the David-stars T$_A$ stacking and (**g**) the T$_A^S$ stacking associated with (**j**) the laddering stack structure in LC-TaS$_2$ crystals verified by HAADF-STEM images collected along (**k**) [100] and (**i**) [$\bar{1}$20] directions. The white vertical lines in panels (**b, k, i**) indicate the *c*-direction. The red solid line in panel (**k**) highlights the atomic misalignment. **h** Schematic top view of monolayer octahedral TaS$_2$. The red and purple arrows indicate the [$\bar{1}$20] and [110] directions, respectively. **e** Schematic energy band diagrams depicting a band-insulator and a Mott-insulator. CB, VB, UHB, and LHB denote the conduction band, valence band, upper Hubbard band, and lower Hubbard band, respectively.

system, making it challenging to reach a definitive conclusion about the insulating nature of 1T-TaS$_2$ crystals.

As the surface-to-volume ratio increases in layered materials, surface and interface effects gain greater significance. The recent discovery of sliding ferroelectricity in two-dimensional (2D) layered homostructures, achieved by controlling the translational degrees of freedom[28–32], naturally prompts a fundamental question: Can stacking modulation be extended to the three-dimensional (3D) counterpart? And how does lattice translation between adjacent layers in 3D layered crystals impact their electronic structures?

In this study, we introduced a laddering stack structure (Fig. 1j) in 1T-TaS$_2$ crystals by deliberately creating fractional misalignment of adjacent layers, leading to a misaligned David-stars T$_A^S$ stacking (Fig. 1g). This unique approach unveiled a distinct insulating state, shedding light on switching between Mott-states and band-insulating states through interlayer stacking and coupling manipulation. With interlayer coupling, the dominance shifts towards the 3D band-insulating states, whereas interlayer decoupling, achieved through the structuring of laddering interlayer sliding, favors the prevalence of 2D Mott-insulating states. This discovery provides compelling evidence of the dualistic insulating nature of 1T-TaS$_2$ crystals. Furthermore, extending this laddering stack structure to other layered materials with similar structural characteristics of intralayer stiffness and interlayer slipperiness opens up exciting possibilities to explore fascinating physics and intriguing properties regardless of their dimensionality.

## Results

### Laddering interlayer sliding and structural characterization

The laddering stack structure is verified using aberration-corrected high-angle annular dark-field scanning transmission electron microscopy (HAADF-STEM), as shown in Fig. 1k. This unique structure features a routine sandwiched unit, whereas a periodically misaligned interlayer stacking arrangement along the out-of-plane *c*-direction (Fig. 1j) distinguishes it from the well-established vertical alignment of Ta-atoms between adjacent layers in 1T-TaS$_2$ (bb-stacking in Fig. 1a, b and Supplementary Fig. 1)[33]. The interlayer sliding occurs along the [$\bar{1}$20] direction within the *ab*-plane guided by red arrows in Fig. 1h, g, as determined from the corresponding HAADF-STEM image (Fig. 1i). Viewing along the [100] direction, a 20-layer periodicity is discerned from a large-scale HAADF-STEM image (see Supplementary Fig. 2). Considering the [$\bar{1}$20] direction interlayer sliding, the Ta-atoms in the upper layer are expected to align with the Ta-atoms in the lower layer after a 40-layer periodicity, corresponding to a translational shift of ~0.015 nm per layer. The 40-layer periodicity is confirmed by the HAADF-STEM image along [110] direction (Supplementary Fig. 3). Here, we name the crystals with octahedral coordinated TaS$_2$ units stacking in a laddering configuration as LC-TaS$_2$.

The interlayer spacing *d* is determined by synchrotron-based X-ray diffraction (XRD) spectroscopy at room temperature (Fig. 1c). The scattering vector *q* for LC-TaS$_2$ is determined to be 10.75 nm$^{-1}$, corresponding to a *d* = 0.584 nm (calculated using *d* = 2π/*q*). This *d* value is slightly smaller than 0.587 nm (*q* = 10.70 nm$^{-1}$) for 1T-TaS$_2$[34], indicating a slight out-of-plane contraction. A shoulder with a lower *q* = 10.69 nm$^{-1}$ originates from residual bb-stacking configurations. It is not surprising that a complete transition to the metastable laddering stack structural phase remains challenging, given that local fluctuation favors the relatively more stable 1T-phase[35,36].

To understand the influence of laddering stack structure on interlayer dimerization, the XRD measurements were performed using Cu K$_\alpha$. Figure 1f displays the superlattice reflections collected at 120 K, and the (00*l*) indexed diffraction peaks are presented in

Supplementary Fig. 4. In the 1T-TaS$_2$ crystal, interlayer dimerization associated with David-stars ordered array results in half-integer (0,0,5/2) and (0,0,7/2) reflections at ~37.9° and ~54.1°, consistent with previous findings[22]. The absence of these superlattice reflections in the LC-TaS$_2$ crystal implies the collapse of interlayer dimerization, either caused by the collapse of CDW states or the absence of a David-stars ordered array.

### Electron localization at the David-star center

To understand the influence of the laddering interlayer sliding on charge redistribution within the David-stars, synchrotron-based photoemission spectroscopy (PES) was conducted at room temperature, where 1T-TaS$_2$ is in the nearly-commensurate CDW (NC-CDW) phase. Within each David-star, there is in-plane electron transfer from outer C-/B-Ta-atoms to the central A-Ta-atoms (Fig. 1d)[37,38]. The distinct core-hole screening effect, resulting from unequal local electron density at specific Ta-sites, leads to further splitting of the Ta $4f_{7/2}$ peak (Fig. 2a). Three components at ~23.64, ~23.10, and ~22.97 eV with an intensity ratio of ~6:6:1 are extracted for 1T-TaS$_2$. The binding energies (BEs) of C-Ta and B-Ta features for LC-TaS$_2$ remain unchanged, indicating the persistence of the CDW states. However, a lower BE shift of the A-Ta feature (~0.05 eV), which is appearance seen in Supplementary Fig. 5, indicates enhanced electron density and localization at A-Ta-atoms. This effect is most likely caused by interlayer decoupling[15], resulting from laddering interlayer sliding.

An alternative explanation for the lower BE feature, attributed to the 2H-TaS$_2$ phase or Ta$^{3+}$ species associated with S-vacancy defects[39], can be ruled out based on the following evidences:

(i) The S $2p$ spectral profile for LC-TaS$_2$ crystals remains almost unchanged (Fig. 2b), indicating the absence of 2H-TaS$_2$. Otherwise, an additional feature at ~160.80 eV would be detected, as we reported previously[39].

(ii) The comparable intensity ratio of Ta/S for both 1T-TaS$_2$ (~0.91) and LC-TaS$_2$ (~0.92) crystals indicates a constant chemical stoichiometry.

### Duality of insulating states

To further understand the influence of the laddering interlayer sliding on the properties of insulating states and CDW order presented at low temperatures, scanning tunneling microscopy/spectroscopy (STM/STS) measurements were conducted at 4.5 K. The STM images of both 1T-TaS$_2$ (Fig. 2d) and LC-TaS$_2$ (Fig. 2e) crystals reveal the well-resolved C-CDW phase accompanied by a commensurate √13 × √13 triangular David-star superlattice, where each bright spot corresponds to one David-star. Remarkably, the long-range CDW order remains intact at the surface of LC-TaS$_2$ crystals. The persistence of David-stars, along with the absence of out-of-plane superlattice reflections (Fig. 1f), implies the absence of out-of-plane ordered array of Davis-stars.

Interestingly, the profiles of the corresponding d$I$/d$V$ spectra change dramatically (Fig. 2c). For the as-grown 1T-TaS$_2$ crystal, two distinct peaks centered at approximately −0.19 and 0.26 eV feature a peak-to-peak energy gap with Δ ~0.45 eV and an edge-to-edge energy gap of ~0.22 eV, consistent with previous experimental results[8,9,11,21,39,40]. However, for the LC-TaS$_2$ crystal, two in-gap states appear at −0.06 eV and 0.18 eV, yielding a narrower energy gap of Δ ~0.24 eV and an edge-to-edge gap of ~0.08 eV. These pronounced differences in edge-to-edge gap features between 1T-TaS$_2$ and LC-TaS$_2$ crystals persist consistently across the surface, as shown in Supplementary Fig. 6d, f. The Δ~0.24 eV closely matches the theoretically predicted Mott-gap of ~0.2 eV for monolayer 1T-TaS$_2$[10], indicating that this insulating state is induced by Mott localization. Nevertheless, this value is smaller than the experimentally observed energy gap of ~0.45 eV in molecular beam epitaxy growth monolayer 1T-TaS$_2$ on a graphene/SiC surface[41], implying a larger screening of Coulomb repulsion in a 3D structure compared to a 2D monolayer. The STS spectra with a narrow energy gap have also been theoretically stimulated and experimentally observed on the cleaved 1T-TaS$_2$ surface with specific David-stars T$_C$ stacking[13,20,21,42], suggesting their common origin of interlayer decoupling. Here, T$_C$ refers to the vertical alignment between A-Ta-atoms and C-Ta-atoms of the adjacent David-stars. These findings imply that the two times larger gap of the as-grown 1T-TaS$_2$ crystal can be attributed to a distinct mechanism, for instance, band-insulating.

### Flat band structure and DFT calculation

To gain insights into the nature of insulating states, the temperature-dependent band structures were investigated by using synchrotron-based angle-resolved PES (ARPES). The results for 1T-TaS$_2$ crystals are

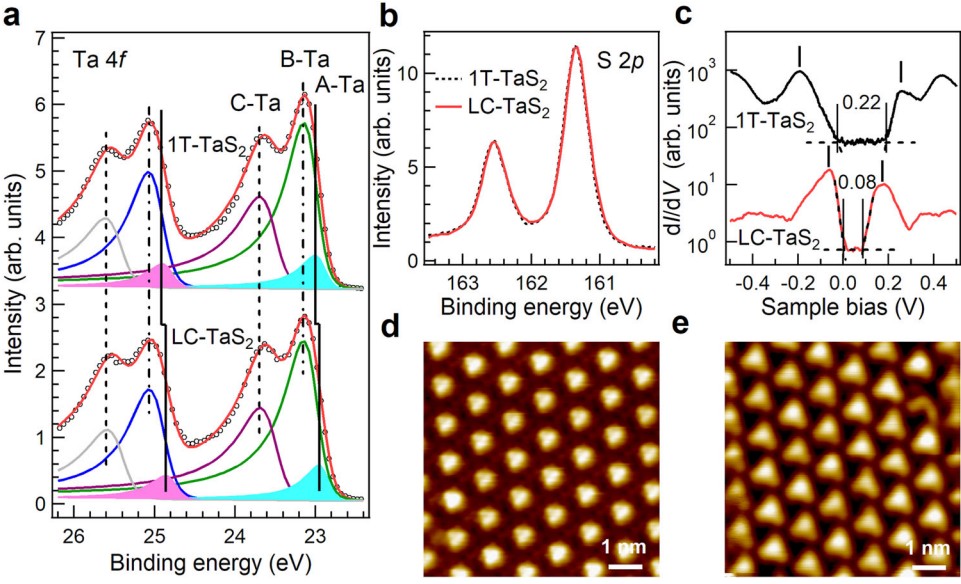

**Fig. 2 | Evolution of chemical composition and electronic structures. a** Ta $4f$ with corresponding fitting curves and (**b**) S $2p$ core-level spectra measured at room temperature using a photon energy of 240 eV. **c** Spatially averaged d$I$/d$V$ spectra with logarithmic intensity scale conducted at David-star centers for 1T-TaS$_2$ and LC-TaS$_2$ crystals at 4.5 K. The curves are vertically shifted for clarity with zero conductance marked by horizontal dashed lines. STM images of the C-CDW phase for (**d**) 1T-TaS$_2$ and (**e**) LC-TaS$_2$ crystals conducted at 4.5 K.

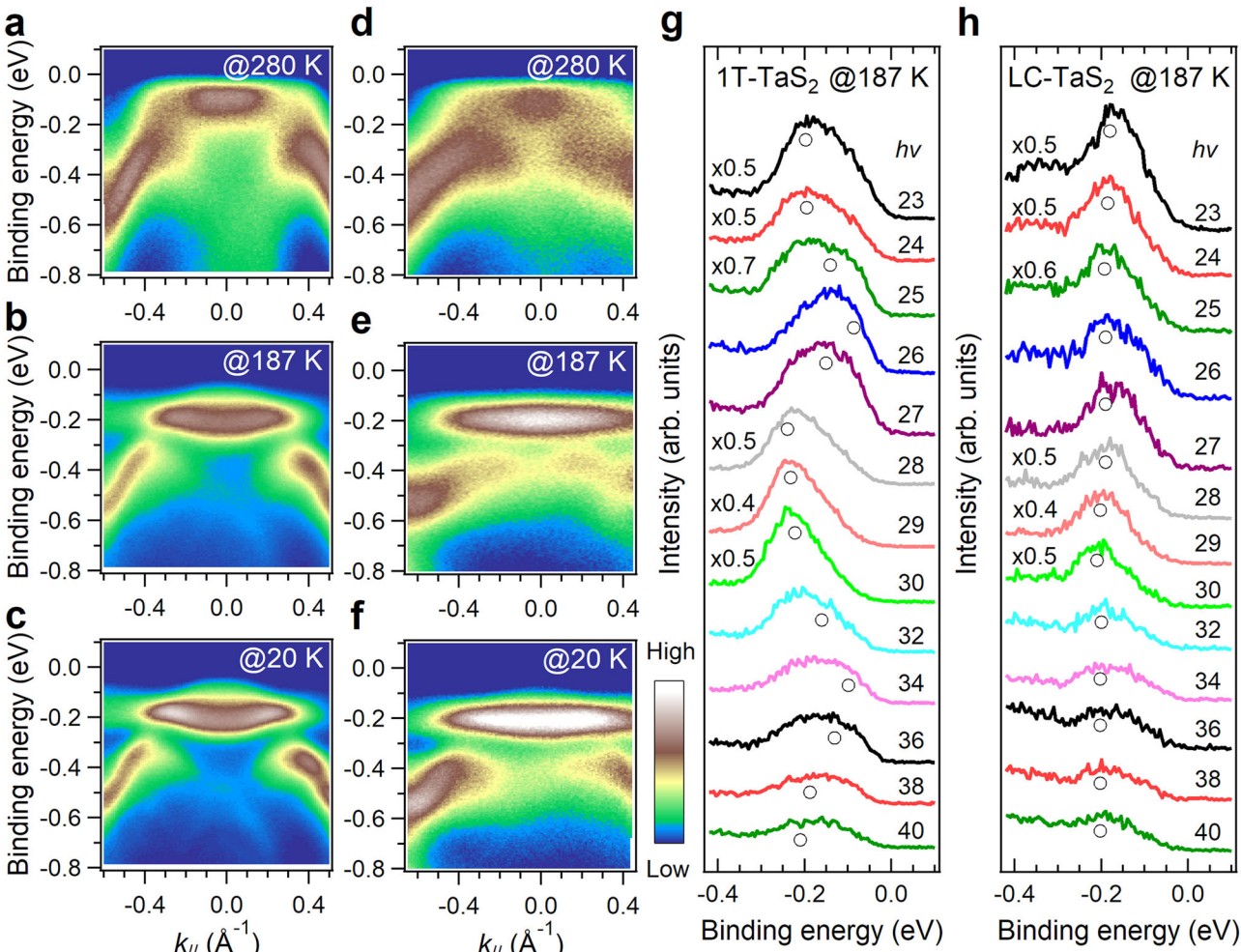

**Fig. 3 | Band dispersions along the $k_{//}$ and $k_z$ direction in different electronic states.** The ARPES data taken along the Γ-M/$k_{//}$ direction (see Supplementary Fig. 7a, b) for (**a**–**c**) 1T-TaS$_2$ and (**d**–**f**) LC-TaS$_2$ at reducing temperatures of 280 K (NC-CDW phase), 187 K (C-CDW phase), and 20 K (insulating phase) with a photon energy of 22 eV. Photon energy dependence of energy distribution curves (EDCs) in the C-CDW phase collected at the Γ-point and 187 K for (**g**) 1T-TaS$_2$ and (**h**) LC-TaS$_2$. The opened circles represent the fitted band positions.

consistent with previous findings[22,43], providing a reliable reference for comparison. In the NC-CDW phase range at 280 K, both 1T-TaS$_2$ and LC-TaS$_2$ exhibit a flat band that emerges near the Fermi level ($E_F$) (Fig. 3a, d). As the temperature decreases to 187 K, the flat bands shift away from $E_F$ and center at ~−0.2 eV (Fig. 3b, e), resulting in an increase in the energy gap.

To investigate the band dispersion along the $k_z$ direction, photon energy-dependent measurements were conducted at 187 K. In 1T-TaS$_2$, the peak position shifts from −0.23 to −0.09 eV (Fig. 3g), whereas for LC-TaS$_2$, the dominant peak at ~−0.2 eV exhibits less photon energy dependence (Fig. 3h). These characteristics were similar to those reported in 1T-TaS$_2$ during a band-to-Mott insulating phase transition triggered by temperature[22]. In contrast, the observed characteristics in LC-TaS$_2$ remain unchanged at 20 K (Fig. 3f and Supplementary Fig. 7d), indicating the insensitivity of the electronic states to temperatures and confirming their intrinsic nature. This feature, along with a distinct interlayer distance difference between LC- and 1T-TaS$_2$ crystals, implies that the periodic laddering stack structure is unlikely to be the structural origin for the heating-triggered surface intermediate Mott-insulator phase[22,44]. However, the possibility of an interlayer atomic sliding confined to the surface region (5–8 layers), such as the irregular sliding observed in layered bulk PbI$_2$[45], cannot be ruled out.

Furthermore, we performed density functional theory (DFT) calculations to gain a comprehensive understanding of electronic

structures in the above STS and ARPES experiments. A four-layer supercell with an ordered David-stars $T_AT_C$ stacking (Fig. 4a), identified by XRD superlattice reflections in 1T-TaS$_2$ (Fig. 1f), was adopted. This configuration has been extensively employed in previous DFT calculations[13,14,20]. The absence of out-of-plane superlattice reflections in LC-TaS$_2$ crystals indicates a disordered David-stars stacking configuration. Aiming to emphasize the influence of atomic layer sliding on the alteration of the electronic structures, $T^S_AT^S_C$ stacking (Fig. 4b), integrating David-stars $T_AT_C$ stacking with a translational shift representing disordered David-stars stacking due to broken translational symmetry along the c-direction, was adopted. The band structure for four layers 1T-TaS$_2$ (Fig. 4d) with David-stars $T_AT_C$ stacking is in good agreement with previous reports[14,15,46]. However, for $T^S_AT^S_C$ stacking with a translational shift of 0.053 nm, the calculated band structures using generalized gradient approximation (GGA) exhibit metallic characteristics with a conduction band crossing the $E_F$ at the L-point (Fig. 4g). To address this discrepancy, the Hubbard $U$ term (GGA + U) is introduced in calculations, which opens a gap (Fig. 4h), confirming its Mott-insulator nature. For comparison purposes, the GGA + U method is also employed to calculate the band structure of $T_AT_C$ stacking, as shown in Fig. 4e. As expected, the limited increase in the energy gap after the introduction of the Hubbard $U$ term indicates that on-site Coulomb repulsion is not the primary factor contributing to the energy gap for $T_AT_C$ stacking (Fig. 4c). In addition, interlayer distance

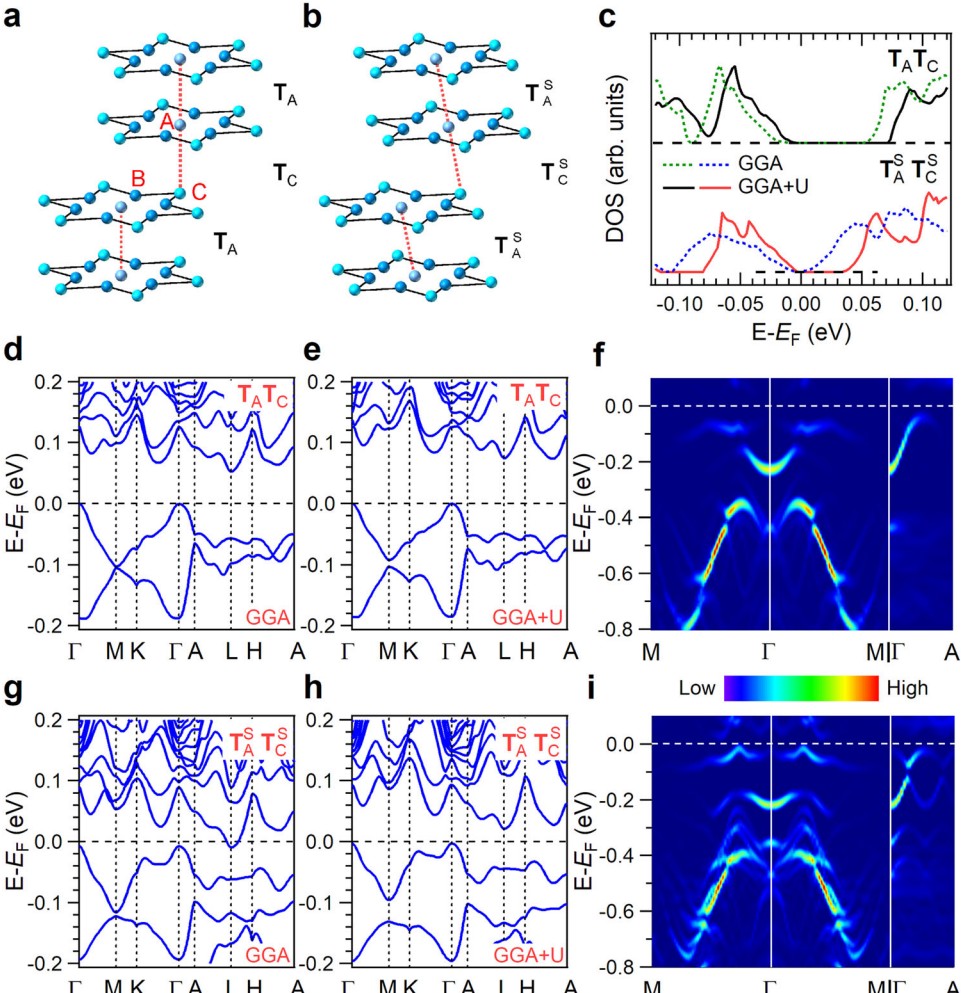

**Fig. 4 | DFT calculated band structures.** Schematic diagrams illustrating (**a**) the $T_AT_C$ and (**b**) $T^S_AT^S_C$ David-stars stacking sequence. **c** Density of States (DOS) for the $T_AT_C$ and $T^S_AT^S_C$ stacking. Band structures near the $E_F$ for $T_AT_C$ stacking calculated using (**d**) generalized gradient approximation (GGA) and (**e**) GGA + U method. Band structure for $T^S_AT^S_C$ stacking using (**g**) GGA and (**h**) GGA + U method. Calculated unfolded band structure for (**f**) $T_AT_C$ and (**i**) $T^S_AT^S_C$ stacking using the GGA + U method.

contraction has negligible influence on the low-energy electronic structure beyond the reduced gap (Supplementary Fig. 9), confirming the dominant roles that laddering stack played in determining the interlayer coupling. The validity of simplifying the system to only 4 layers and employing a larger translational shift is discussed in the Methods part.

The density of states (DOS) represents gap opening induced by Mott-localization for the laddering stack structure (Fig. 4c). Although GGA calculations typically underestimate the gap size[13], the DOS calculations show that the gap closes after laddering interlayer sliding, and reopens with a smaller value after introducing the Hubbard $U$ term. This trend is consistent with the STS results (Fig. 2c).

The unfolded bands in the unit-cell Brillouin zone demonstrate the reduction in band dispersion along both Γ-A and Γ-M directions. Specifically, the dispersion of the flat band near −0.2 eV for $T^S_AT^S_C$ stacking (Fig. 4i) becomes flatter than that for $T_AT_C$ stacking (Fig. 4f). This is consistent with ARPES results at -187 K, where the flat band centered at ~−0.2 eV for LC-TaS$_2$ (Fig. 3e) is much narrower and less dispersive than that of 1T-TaS$_2$ (Fig. 3b). In addition, the dispersion along the Γ-A direction is reduced to -0.10 eV for $T^S_AT^S_C$ stacking from -0.18 eV for $T_AT_C$ stacking. The reduced band dispersion aligns with observations from ARPES results. For LC-TaS$_2$, the dominant peak at ~−0.2 eV exhibits less photon energy dependence (Fig. 3h).

The ARPES spectra, with a detected depth of 2–3 nm for electrons with 18 eV kinetic energy, extend across three layers[47], providing multilayer information rather than characteristics of the top surface layer[22]. In LC-TaS$_2$, the 2D flat bands close to $E_F$ along both the Γ-A/$k_z$ and Γ-M/$k_{//}$ directions in the ARPES measurements and the wide energy gap determined from the STS spectra, in agreement with DFT calculations, confirms its 2D Mott-insulator nature. On the other hand, significant band dispersion in 1T-TaS$_2$ implies its classification as a 3D band insulator[22]. It is noteworthy that, although DFT calculations based on $T^S_AT^S_C$ stacking generally capture the reduced band dispersion character in LC-TaS$_2$, deviations in actual band dispersion values are present. This discrepancy may arise from alterations in the David-stars stacking sequence induced by atomic layer sliding, as illustrated in Supplementary Fig. 8, warranting further investigation.

**The effects of laddering interlayer sliding on electrical transport**
To explore the distinct macroscopic properties associated with this kind of duality of insulating states, anisotropic electrical transport measurements were conducted. Figure 5a, b show the temperature dependence of the mobility $\mu$ and carrier concentration $n$, respectively. The values were calculated using the formulas $n = (eR_H)^{-1}$ and $\mu = R_H/\rho_{xx}$, where $\rho_{xx}$ and $R_H$ represent resistivity and Hall coefficient, respectively, as shown in Supplementary Fig. 12. The negative and

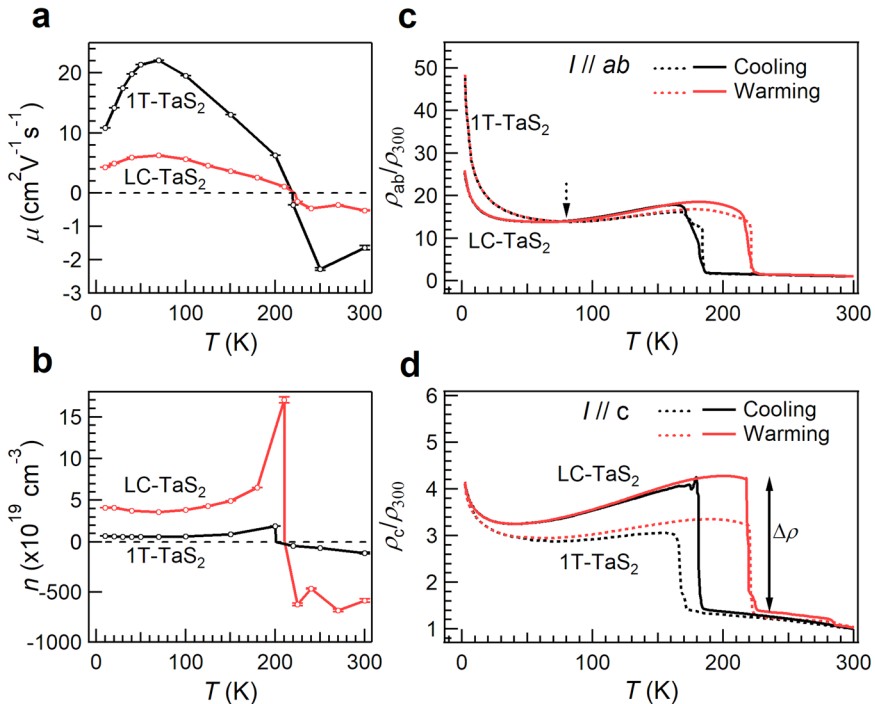

**Fig. 5 | Anisotropic electrical transport properties. a** The carrier mobility $\mu$ and (**b**) carrier concentrations $n$ determined from the Hall measurements (see Supplementary Fig. 12). The negative values in panels (**a, b**) correspond to the electron concentration and electron mobility, respectively. The error bars are derived from the standard deviation of fitting for Hall resistivity, as shown in Supplementary Fig. 12d, e. Temperature dependence of (**c**) in-plane $\rho_{ab}/\rho_{300}$ and (**d**) out-of-plane $\rho_c/\rho_{300}$ for 1T-TaS$_2$ and LC-TaS$_2$ upon cooling and warming.

positive signs denote the electron and hole carrier types, respectively. The obtained parameters for 1T-TaS$_2$ are consistent well with the literature[48]. The sign change for both parameters corresponds to the transition from NC-CDW to C-CDW phase, as well as the gap opening observed in ARPES results (Fig. 3). In LC-TaS$_2$, the electron localization and increased effective mass arising from the narrowing of flat bands, as revealed by ARPES, leads to a reduction in $\mu$ to -0.3 times of the value of 1T-TaS$_2$. Meanwhile, the ~6-fold increase in $n$ for LC-TaS$_2$ is dominated by the narrower Mott-gap, as revealed by STS, although the increase in the effective mass of carriers can offset part of the contribution. It is clear that the relatively large $\mu$ and small $n$ observed in intrinsic 1T-TaS$_2$ crystals, are conversed into the lowered $\mu$ and elevated $n$ in LC-TaS$_2$.

To better understand the underlying physics of laddering interlayer sliding, we normalized the resistivity to the values of NC-CDW state at 300 K ($\rho_{300\,K}$). This helps to exclude the effects of extrinsic factors and facilitate a comparison with 1T-TaS$_2$ (Fig. 5c). Above 80 K, the resistivity behavior remains consistent even after laddering interlayer sliding, indicating unchanged in-plane David-stars configuration, although both $\mu$ and $n$ are modulated by interlayer coupling. Below 80 K, the weaker insulating behavior of LC-TaS$_2$ is consistent with its narrower energy gap.

For electrical transport between layers, resistivity with $I//c$ was measured and plotted in Fig. 5d. The abrupt resistance jump ($\Delta\rho$) induced by the first-order NC-to-C-CDW phase transition in $\rho_c/\rho_{c(300\,K)}$ for LC-TaS$_2$ is much larger than that for 1T-TaS$_2$. This significant increase of $\Delta\rho$ can be attributed to the reduced band dispersion along $k_z$ and, consequently, a larger effective mass of carriers in LC-TaS$_2$ rising from the Mott-insulating mechanism.

## Discussion
Upon reviewing the aforementioned results, in $T_A$ stacking (Fig. 1d), the David-stars with unpaired electrons dimerize to form bilayers with an even number of electrons[8,10,11,14–16,20–22,49]. Conversely, in the fractional translation of $T^S_A$ stacking (Fig. 1g), the non-dimerization of David-stars preserves an odd number of electrons within each David-star. This results in an enhancement of electron density within the David-star, consequently strengthening electron-electron interactions[9,19,20,50,51]. Here, the laddering stack structure is expected to gradually weaken the interlayer coupling by reducing the out-of-plane wavefunction overlap integral between David-stars of adjacent layers, thus reducing out-of-plane bandwidth $t_\perp$. This reduction in $t_\perp$ leads to an increase in Coulomb repulsion $U$ due to the reduced electron screening. Consequently, the increased $U/t_\perp$ ratio explains the switching from a band-insulator to a Mott-insulator mechanism[52].

Our discovery introduces a translational degree of freedom for precise tuning of bulk crystal properties. It stands out for its simplicity, cleanliness, and seamless integration into device fabrication processes. This method opens up opportunities to investigate low-dimensional physics in layered 3D crystals[53], expanding its applications beyond the realm of sliding ferroelectricity[28,29,32].

## Methods
### Samples Preparation
High-quality 1T-TaS$_2$ single crystals were grown by a chemical vapor transport (CVT) method[34,54,55]. First, tantalum and sulfur power with a 1:2 molar ratio were mixed and vacuum-sealed in the quartz tubes. After annealing at 800 °C for 3 days, polycrystalline TaS$_2$ was obtained by quenching the quartz tubes in an ice-water mixture. Subsequently, the polycrystalline TaS$_2$ powder and iodine (as a transport agent) were vacuum-sealed in quartz tubes. The quartz tubes were heated in a two-zone furnace for 10 days with different temperatures in the hot zone (850 °C) and cold zone (750 °C). The 1T-TaS$_2$ crystals were obtained after quenching the quartz tubes in an ice-water mixture. The obtained 1T-TaS$_2$ crystals were transferred to a vacuum chamber (better than $10^{-2}$ mbar) and annealed at 150–200 °C for 3–4 h. The LC-TaS$_2$ crystals are obtained after cooling to room temperature naturally. All collected crystals were stored in an Ar-protected glove box before further

characterization. Subsequently, the crystals were cut to pieces for electron microscopy and (synchrotron-based) XRD and PES characterization.

## Electron microscopy

HAADF-STEM images of the cross-sectional samples were collected at room temperature on an FEI Themis Z aberration-corrected scanning transmission electron microscope operating at 300 keV with a convergence angle of 30 mrad[56,57].

## XRD measurement

The synchrotron-based XRD measurements were performed at the BL14B1 beamline of the Shanghai Synchrotron Radiation Facility (SSRF) using X-rays with a wavelength of $\lambda = 1.24$ Å and grazing incident angles of 1.0°. The corresponding spectra collected at room temperature are presented in scattering vector $q$ coordinates by using the equation $q = 4\pi\sin\theta/\lambda$, where $\theta$ is half of the diffraction angle. The $q$ has been calibrated by measuring the synchrotron-based XRD of a lanthanum hexaboride reference sample.

Lab-based XRD characterizations were performed on a Bruker D8 diffractometer using Cu K$_\alpha$ radiation to assess the quality of the obtained crystals. XRD data were also collected on 1T- and LC-TaS$_2$ crystals after cooling down to 120 K.

## Synchrotron-based PES and ARPES

High-resolution PES was collected at the BL11U beamline of the National Synchrotron Radiation Laboratory (NSRL, China). The single crystals were cleaved and transferred into an ultra-high vacuum (UHV) chamber with a base pressure of $1 \times 10^{-10}$ mbar. The S 2$p$ and Ta 4$f$ spectra were measured at normal emission and room temperature using 240 eV photon energy. The photon energy was calibrated using the Au 4$f_{7/2}$ core-level peak (84.0 eV) of a gold foil in electrical contact with the samples. The least-squares peak fitting was performed employing a Shirley background and asymmetric peak profiles for TaS$_2$ species.

The ARPES measurements were performed at the BL13U beamline of NSRL. The single crystals, previously qualified by XRD, were cooled to 280 K, cleaved, and measured in situ in a UHV chamber with a base pressure of $1 \times 10^{-10}$ mbar. The crystals were further cooled to a selective temperature and maintained for 10 min before measurement. The Fermi energy was referred to as a gold foil in electrical contact with the samples.

## STM/STS measurement

The STM/STS measurements were conducted in an Omicron LT-STM system. The obtained 1T- and LC-TaS$_2$ single crystals, qualified by XRD, were cleaved and transferred into a UHV prep-chamber with a base pressure of $5 \times 10^{-10}$ mbar. After degassing at approximately 100 °C for 1 h, the distinct samples were transferred to the analysis chamber with a base pressure of $1 \times 10^{-10}$ mbar for STM/STS measurements. STM topography images were collected in constant-current mode. STS measurements were performed in constant-height mode using a standard lock-in technique ($f = 773$ Hz, $V_{r.m.s.} = 15$ mV). The STM tip was calibrated spectroscopically on the Au(111) surface. All data were collected at 4.5 K.

## Transport measurement and sources of error in data fitting

The electrical transport measurements for freshly cleaved single crystals, cut from XRD-qualified samples, were carried out by using a physical property measurement system (PPMS, Quantum Design) with a standard six-probe method. The warming and cooling rates are 3 K/min. The in-plane and out-of-plane resistivity data for LC-TaS$_2$ are collected on two identical samples cut from one single crystal.

While linear fittings to Hall resistivity provide satisfactory descriptions of experimental data (Supplementary Fig. 12d, e),

standard fitting errors persist. These errors subsequently introduced inaccuracies in related parameters, such as $\mu$ and $n$ (Fig. 5a, b). Notably, the standard fitting errors for Hall coefficient were four orders of magnitudes smaller than the obtained parameters, ensuring the high reliability of both the fitting process and its related outcomes.

## DFT calculation

The first-principles calculations based on DFT were performed using the Vienna Ab initio Simulation Package (VASP)[58,59]. The pseudopotential was described using the projector augmented wave (PAW) methods, and the exchange-correlation interaction was treated using GGA, which is parameterized by Perdew-Burke-Ernzerhof (PBE)[60]. The on-site Coulomb repulsion $U = 2.94$ eV, previously established using a self-consistent method[61], was included for the tantalum 5$d$ orbitals. The energy cutoff for the plane wave was set to 400 eV. The van der Waals interaction was accounted for using Grimme's DFT-D3 of the semi-empirical method[62]. Brillouin zone sampling was performed using a $5 \times 5 \times 5$ $k$-point mesh for electronic properties calculation. The band unfolding was processed using the VASPKIT code[63].

To examine the influence of laddering interlayer sliding on the electronic structure, first-principles calculations were performed. The initial assumption was made for pristine 1T-TaS$_2$ crystals, which were considered to exhibit an out-of-plane paired T$_A$T$_C$ David-stars stacking sequence (Fig. 4a). This specific stacking arrangement, characterized by stacking vectors of $\mathbf{T}_A = \mathbf{c}$ and $\mathbf{T}_C = 2\mathbf{a} + \mathbf{c}$, has been proposed in previous experimental studies and was employed in DFT calculations[13,14,20]. To simulate the interlayer sliding effect, the corresponding T$^S_A$T$^S_C$ stacking configuration was considered (Fig. 4e). In this configuration, $\mathbf{T}^S_A = \delta\mathbf{a} + \mathbf{c}$ and $\mathbf{T}^S_C = (2+\delta)\mathbf{a} + \mathbf{c}$, with $\delta\mathbf{a}$ representing the lateral sliding distance.

Simulating the experimentally observed 40-layer supercell structure, which contains 1560 Ta- and S-atoms, poses challenges. As a result, a simplified structure with a minimum of 4 layers or one T$^S_A$T$^S_C$ stacking period was adopted. To compensate for the effects of this structural simplification, a larger sliding distance $\delta\mathbf{a}$ of 0.053 nm (1/6$\mathbf{a}$) was used. The validity of this approach was verified by varying $\delta\mathbf{a}$ from 0.039 nm (1/8$\mathbf{a}$) to 0.106 nm (1/3$\mathbf{a}$). Supplementary Fig. 10 demonstrates that the band structures calculated with $\delta\mathbf{a} \geq 0.053$ nm (1/6$\mathbf{a}$) accurately reproduce results from the four-layer supercell structure. To clarify the coordination effect between supercell layer numbers and sliding distance in representing experimental observation, extended calculations of eight-layer supercell structures were performed. Supplementary Fig. 11 illustrates the corresponding band structures for various $\delta\mathbf{a}$. Notably, a 0.039 nm (1/8$\mathbf{a}$) value is identified to close the gap for a larger eight-layer supercell structure, which is smaller than 0.053 nm (1/6$\mathbf{a}$) for the four-layer supercell structure (Supplementary Fig. 11e). The observations lead to the conclusion that a sliding distance $\leq\sqrt{3}/n\mathbf{a}$ can effectively close the gap for an $n$-layer supercell structure (Supplementary Fig. 11f). Therefore, it is reasonable to propose that $\sqrt{3}/40\mathbf{a}$ sliding can close the gap in a 40-layer supercell structure, as observed experimentally.

Notably, the GGA + U method has limitations regarding the accuracy of gap value in this system, given its sensitivity to the on-site $U$ value. A potential alternative approach is the DFT + GOU method, where the self-consistent $\bar{U}$ acts on the entire David-star cluster. This method may offer a more precise description of the electronic structure[19,42], warranting further theoretical investigation.

## Data availability

All data generated in this study are provided in the Article and the Supplementary Information. Additional data related to this work are available from the corresponding author upon request. Source data are provided with this paper.

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

## Acknowledgements
The authors thank the BL11U and BL13U beamlines of the National Synchrotron Radiation Laboratory (NSRL), and the BL14B1 and BL02U2 beamlines of the Shanghai Synchrotron Radiation Facility (SSRF) for providing the beam time. A portion of this work was supported by the High Magnetic Field Laboratory of Anhui Province. The authors thank Prof. Dongsheng Song from Anhui University and Prof. Haifeng Du from the High Magnetic Field Laboratory of the Chinese Academy of Sciences for their valuable discussions.This work was supported by the National Key Research and Development Program of China (Grant Nos. 2021YFA1600200 (Y.S. and X.L.), 2019YFA0405601 (J.Z.), 2022YFA1403203 (W.L.), and 2023YFA1607402 (X.L.)), and the Innovation Program for Quantum Science and Technology (Grant No. 2021ZD0302802 (Y.X. and Z.S.)), and the National Natural Science Foundation of China (NSFC) (Grant Nos. 12074385 (Y.H.), 12074372 (H.X.), 12204486 (Y.W.), 11727902 (H.X.), 11674326 (Y.S.), 11474288 (Y.X.), 12274412 (Y.S.), U1832141 (X.L.), U1932217 (X.L.), U2032215 (Y.S., W.L.), and 22272157 (J.Z.)). Y.X. is supported by Anhui University through the start-up project (Project No. S020318001/020).

## Author contributions
L.C. H.X., Y.X., W.L., and Z.S. conceived and supervised this project. Z.L., X.L., and J.G. prepared the single crystals. Y.W. and Y.H. performed the transport measurements. J.J. and J.T. prepared TEM samples and recorded STEM images. Z.L. and H.J. performed the PES studies. T.L. and S.C. collected the ARPES data. Y.W. and Y.Y. took the XRD measurements. R.L. and W.L. performed DFT calculations. X.L., W.L., Y.S., J.Z., X.G., and Z.S. analyzed the data. Y.W., Z.S., H.X., Y.X., W.L., and L.C. drafted the manuscript. All authors contributed to manuscript development and have given approval to the final version of the manuscript.

## Competing interests
The authors declare no competing interests.
