## [Peer Review File · Nature Communications]

Editorial Note: Parts of this peer review file have been redacted as indicated to maintain the confidentiality unpublished data.REVIEWER COMMENTS

Reviewer #1 (Remarks to the Author):

In this paper, Wang and colleagues describe HAADF-STEM, XRD, XPS, STM and ARPES measurements on 1T-TaS₂ and samples with an apparent tilted or 'ladder' stacking structure that they describe as '20T-TaS₂'. They perform DFT calculations of bilayer-stacked 1T-TaS₂ with a small layer-by-layer in-plane shift (less than one lattice constant per layer) to simulate the ladder stacking, and they use this to help understand the properties of the supposed 20T-TaS₂ samples. Finally, they present in-plane and out-of-plane transport measurements for each kind of sample.

I think each of the measurements or calculations in the paper are of good quality on their own.

The takeaway message of the paper seems to be that in 'ladder-stacked' 20T-TaS₂, the star-of-David clusters do not undergo layer-dimerization, so that their insulating behavior must be ascribed to the Mott mechanism and not the band insulator mechanism. To support this claim, the Authors point to an observed reduction of the insulating gap, from the larger band gap of the layer-dimerized structure to the smaller Mott gap of the layer-decoupled structure.

Taken at face value I think a general claim that a macroscopic band-to-Mott transition of the insulating state, controlled by some inter-layer degree of freedom, would be impactful enough to warrant publication in Nature Communications. However, I have serious doubts about several aspects of the work, and I believe that the following queries must be thoroughly answered before any recommendation can be made:

1)

First, the Authors do not clearly describe how they created the samples that they call '20T-TaS₂'. They mention in the Methods section that they annealed the crystals after growth, but they do not actually attribute the transformation of 1T-TaS₂ into 20T-TaS₂ to this annealing process. At the very least, in order for the main results of the paper to be reproducible by other groups, the Authors should explain how they were able to synthesize 20T-TaS₂ crystals. Can the Authors demonstrate that the procedure is reproducible? How many of each kind of sample (1T-TaS₂ and 20T-TaS₂) were created for this work?

2)

This may be the most important query: The structure that is introduced here, namely the 20T structure, is inferred from only one HAADF-STEM image with a tiny field-of-view. But it is unclear how the Authors know that the other measurements (STM and ARPES) are also performed on a 20T-structured region of the sample. It seems necessary to assume that the 20T structure is uniform throughout the whole of some crystals, and totally absent in other crystals, and that the Authors can use some method (e.g. XRD) to sort these crystals from each other. The justification for these assumptions needs to be demonstrated. Can the Authors demonstrate that the 20T structure can be uniformly observed throughout multiple parts of the same sample?

3)

Can the Authors clearly demonstrate that the apparent shear seen in the HAADF-STM image (Fig. S1 for example) really corresponds to a 'ladder stacking' in the sample, and not a shear distortion due to the properties or calibration of the microscope?

4)

The stacking displacement between two layers is a 3D vector, which has two in-plane components. To establish both of these in-plane components, it is necessary to take at least two HAADF-STEM images of the same sample from different directions, e.g. one along the (100) direction and one along the (010) direction. Can the Authors do this? If not, how do we know what the in-plane stacking displacement is?

Also, how do the Authors know that the sample that they call 1T-TaS₂ does **not** have a shear distortion if they have only viewed it from one direction?

5)

The Authors note that the STM conductance spectra shown in Figure 2(c) likely corresponds to the so-called T_A-stacked and T_C-stacked terminations of the layer stacking pattern of the CDW [described in Ref. 20]. But the figure strongly suggests that the difference between the two conductance curves can be attributed to the difference between the 1T and 20T structures. This is misleading. If the Authors want to make a comparison between STM measurements on the 1T and 20T samples, I think they should find the directly comparable surfaces of each case, i.e. the T_A surface of 1T and 20T samples for comparison (and likewise, the T_C surface of 1T and 20T samples). I think this would allow a proper comparison to the DFT and ARPES results.

6)

Somewhat related to the point above: Is it possible that the difference between the ARPES results shown in Figs. 3(a-c) and Figs. 3(d-f), and also between Figs. 3(g) and 3(h), could correspond to observations of the T_A-stacked and T_C-stacked surface layers, and not to the observation of the 1T and 20T structures?

7)

The Authors perform DFT calculations to anticipate the electronic properties of the 20T structure. They include an in-plane displacement of 0.39 Angstroms or more and observe that as the displacement increases, the gap gradually closes. (They show that the gap remains open if the displacement is only 0.39 Angstroms.) The in-plane displacement in the putative 20T-TaS₂ sample would be only about 0.17 Angstroms, and it seems clear from the trend shown in Fig. S5 that there will be a gap, *without* the Hubbard U, for a displacement of 0.17 Angstroms.

8)

In the DFT calculations modelling the 1T-TaS₂ case [e.g. Fig. 4(d)], why do the Authors only show GGA results and not GGA+U? The on-site Coulomb repulsion must still exist in the layer-dimerized band insulating 1T-TaS₂, even though it is not the primary reason for the band gap to appear.

9)

Related to the point above, the Authors present a comparison between GGA band structure for the 1T structure, and GGA+U band structure for the 20T structure [Figs. 4(a) and 4(h)], thereby making a comparison when two variables are changed simultaneously. This is confusing. To me it seems that the GGA+U results for each case should be compared.

10)

I fail to understand why the Authors did not simply perform the DFT calculation with an in-plane shift of about 0.17 Angstroms, directly corresponding to the 20T-TaS₂ structure. I understand that the Authors can refer to multiple values of in-plane displacement to emphasize that it may be a control parameter to adjust electronic structure in what they call 'slidetrionics', but at the very minimum it is necessary to present a DFT treatment that corresponds exactly to the sample that is observed in experiment, to establish that the basic idea is valid.

11)

A minor query: The Authors interpret the XRD as indicating that the inter-layer spacing is reduced for the 20T structure, but they interpret the XPS as indicating that the 20T structure has relatively decoupled layers (line 119). Are these interpretations consistent with each other?

Reviewer #2 (Remarks to the Author):

Wu et al. report on a new structural modification that they found in the well studied layered van der Waals material 1T-TaS₂. They show that annealing a 1T-TaS₂ single crystal followed by cooling to room temperature creates a structure in which the layers are slightly shifted with respect to each other yielding a laddering stack structure which the authors call 20T-TaS₂ as the period of this structure along the c-axis is 20 layers. The authors then employ a comprehensive combination of (angle resolved) photo-emission spectroscopy (AR)PES, scanning tunneling spectroscopy (STS), x-ray diffraction (XRD) and density functional theory (DFT) to study the electronic structure of 20T-TaS₂ in comparison to 1T-TaS₂. The two main conclusions of this study are, firstly, that the well-known star-of-David (SOD) charge density wave (CDW) structure in 20T-TaS₂ is essentially the same as in regular 1T-TaS₂. Secondly, they conclude that due to the interlayer shifts in 20T-TaS₂ the interlayer hybridization between SODs in adjacent layers is effectively reduced which in turn gives rise to increases electron-electron correlations rendering 20T-TaS₂ as being a Mott-insulator.

The manuscript is well written and accessible to non-expert readers. While I find the discovery of the laddering stack structure 20T-TaS₂ certainly very interesting, I have major concerns about the robustness of the drawn conclusions as explained in detail below:

1) My first point concerns the robustness of the assumption that the TaTc CDW stacking is indeed preserved in 20T-TaS₂. The STS and PES data nicely indicate that the SOD structure of the commensurate CDW is preserved at the surface of 20T-TaS₂. I think this is a rather robust result but maybe not utterly surprising as the surface layer is less effected by the sliding distortion than a layer deep in the bulk. However, it is well-known that the stacking order of the CDW itself very strongly effects the electronic structure. The authors assume that the alternating TaTc stacking is preserved and not effected by the sliding distortion. As far as I can see there is no experimental data to backup this assumption. It would be very interesting for instance to see the XRD superlattice reflections and in particular their peak-profiles in the direction perpendicular to the layers which directly reveals the CDW stacking. I am wondering if the GIXRD data perhaps already contain this information.

2) My next point concerns the DFT results: It is not clear to me how the band structure shown in Fig 4 (b) and the unfolded band structure in Fig 4 (d) are related? In Fig 4 (b) there is a hole like band at the Gamma point which is completely absent in the unfolded band structure. Likewise the band

between Gamma and A looks very different in Fig 4(b) and (d). The same issues occur for Fig 4 (f) and (h). Are these band structures really calculated using the same crystal structure as the text and caption suggests? While the unfolded band structures in Fig 4 (d) and (h) are more or less consistent with the literature, I am not aware that this hole like band at the Gamma point which disperses to -0.1 eV at the M point has been reported before for the TaTc stacking with periodic boundary conditions. I think the authors need to revisit their calculations or better explain to which crystal structures they belong and how they relate to previously published electronic structures.

3) Even if the DFT calculations are correct, which seems to be the case at least for the unfolded band structures in Fig. 4 (d) and (h), the agreement with the ARPES data is, in my opinion, not as good as the authors claim. For instance the band at the Gamma point at -0.2eV in Fig 3 (e,f) is rather flat instead of electron like as the DFT calculation in Fig 4 (h) predicts. Perhaps some EDC could help to clarify this point more clearly. The electron-pocket like shape at Gamma is indeed a prominent characteristic of the TaTc stacking and therefore I believe it could well be that due to the sliding distortion a different stacking order occurs. Again, XRD could help to clarify this question.

4) Also concerning the agreement between ARPES and DFT, I noticed that the k_z dispersion in the APRES data in Fig 3. (h)

is nearly zero. The authors claim in line 204ff that this is in accordance with their DFT calculations. However, these calculations would predict a dispersion of roughly 0.1eV.

5) In line 231ff, a connection is made between the charge excitation gap and the transport properties at low temperatures. While this is correct in principle, it should be noted that the gaps derived from the Arrhenius diagram of electrical resistance are of the order of a few meV and thus much smaller than the gaps calculated in GGA+U or seen in STS. In general, the traditional GGA+U method may not be a good approach to describe the electronic structure of this compound, as it may be better described as a cluster Mott system, where the Mott-Hubbard U acts on the whole SOD cluster rather than on the single Ta site. Such calculations have been reported here, for example:
<https://iopscience.iop.org/article/10.1088/2053-1583/ace374/meta>

6) In the Discussion the authors conclude that the laddering stack structure weakens the interlayer coupling by reducing the wave function overlap in adjacent layers. However, in addition to the the in-plane shift of $\sim 0.17\text{\AA}$, the XRD data indicate a reduction of the c-lattice parameter of about 0.03\AA . Was this shortening of the c-lattice parameter taken into account in the DFT calculations? I could imagine that this c-lattice parameter contraction, even if relatively small, may have a significant impact on the low-energy electronic structure. In this context, it is also important to note that even a relatively low hydrostatic pressure completely suppresses the CCDW phase and its characteristic TaTc stacking in favor of the NC-CDW phase in 1T-TaS₂, perhaps mainly due to the compression of the c-

lattice parameter, since the lattice is much stiffer in the ab plane than along c. Accordingly, the TaTc-CDW stacking of the CCDW phase is very sensitive to changes in the interlayer spacing.

7) The authors should definitely provide more experimental details about the annealing procedure to produce the 20T-TaS₂ structure to ensure reproducibility.

To summarize, I think the discovery of the 20T TaS₂ structure is very interesting. If the authors were to describe the annealing protocol in more detail so that their results could be reproduced by others, I think it would certainly be worth publishing. However, the conclusion they have drawn regarding the persistence of TaTc stacking in 20T-TaS₂ and the strengthening of electron-electron correlations is not very solid in my opinion, and I cannot recommend publishing the manuscript in its current form.

Reviewer #3 (Remarks to the Author):

Yihao Wang et al. report the different structural and electronic properties of the 1T-TaS₂ and 20T-TaS₂. The latter has an unconventional stacking order, in which the David's stars are slightly displaced with respect to the 1T structure. An impressive amount of different experimental methods are employed in order to characterize the samples as: STEM, XRD, STM, ARPES, Transport. By these means the authors convincingly show that the 1T-TaS₂ phase corresponds to a dimerized band insulator whereas the 20T-TaS₂ is an undimerized Mott-insulator. The outcome is important and resolves controversial questions on a compound which is currently attracting strong interest. I am confident that the reported results will impact a broad community. The article is clearly written and most of it can be easily understood by non-specialists. Therefore, I recommend publication in Nature Communication.

In the following I report some remarks that the authors should consider before publication:

1) The authors say "This value closely matches the theoretically predicted Mott-gap of ~ 0.2 eV for monolayer 1T-TaS₂". In reality, the STM of single layer 1T-TaS₂ has been already measured, and shows a much larger gap than 0.2 eV (H. Lin et al., Nano Research 13, 133 (2020)). It is no surprise that there is no agreement between 20T-TaS₂ and the monolayer, mainly because of the screening of Coulomb repulsion in a 3D structure. The authors should instead consider to cite the article: J. Dong et al., 2D Mater. 10, 045001 (2023), where ab-initio calculations with self-consistent screened U are calculated both for a 3D dimerized and 3D undimerized structure. The DOS of these calculations is in

good agreement with the STM data measured by the authors for the dimerized 1T-TaS₂ and undimerized 20T-TaS₂.

2) Not being an expert in transport measurement, I do not understand the meaning of a negative carrier density and mobility in Fig. 5a,b). Moreover, it is not explained how the authors extract these numbers. Via Hall effect? Please clarify this point.

3) It is surprising that stacking disorder does not scramble a sliding occurring over 20 unit cells in the 20T structure. Moreover, the rapid quench to obtain 1T-TaS₂ is generally done to avoid the transformation in 2H polytype. How does it come that unquenched TaS₂ acquires the 20T structure instead of the 2H one?

4) The authors do not talk about repeatability of the experiment. Are the samples from the same batch always showing the similar properties? Is the STEM looking the same after every cut? Does the STM depend on the sample position (this is indeed often reported!)?

5) In the temperature dependent ARPES data, I expect the 1T-TaS₂ to show a transition to the 20T-TaS₂ upon heating and just before entering the NC CDW phase (see ref. 22 of your manuscript and J. Dong et al., Physical Rev. B 108, 155145 (2023)). Is this happening?

General response:

In response to reviewer comments and to provide a more accurate description of the laddering stack behaviors in our experiment, we have replaced the term '20T-TaS₂' with 'LC-TaS₂ (octahedral coordinated TaS₂ units stacking in a laddering configuration)' in the revised manuscript. This adjustment aims to elucidate all reviewer comments effectively and enhance reader comprehension.

Point-by-point responses to Reviewers of manuscript NCOMMS-23-42393-T

Comments in black –Author reply in blue –Revisions in purple with yellow highlight.

Reviewer #1 (Remarks to the Author):

In this paper, Wang and colleagues describe HAADF-STEM, XRD, XPS, STM and ARPES measurements on 1T-TaS₂ and samples with an apparent tilted or 'ladder' stacking structure that they describe as '20T-TaS₂'. They perform DFT calculations of bilayer-stacked 1T-TaS₂ with a small layer-by-layer in-plane shift (less than one lattice constant per layer) to simulate the ladder stacking, and they use this to help understand the properties of the supposed 20T-TaS₂ samples. Finally, they present in-plane and out-of-plane transport measurements for each kind of sample.

I think each of the measurements or calculations in the paper are of good quality on their own.

The takeaway message of the paper seems to be that in 'ladder-stacked' 20T-TaS₂, the star-of-David clusters do not undergo layer-dimerization, so that their insulating behavior must be ascribed to the Mott mechanism and not the band insulator mechanism. To support this claim, the Authors point to an observed reduction of the insulating gap, from the larger band gap of the layer-dimerized structure to the smaller Mott gap of the layer-decoupled structure.

Taken at face value I think a general claim that a macroscopic band-to-Mott transition of the insulating state, controlled by some inter-layer degree of freedom, would be impactful enough to warrant publication in Nature Communications.

Author reply:

We appreciate thorough review provided by Reviewer on our manuscript and the positive evaluation of the individual measurements and calculations detailed in this paper.

The succinct summary precisely captures the essence of our study. We are particularly encouraged by your suggestion that our research could merit publication in *Nature Communications*.

We are fully committed to addressing the suggestions and refining our manuscript accordingly. The constructive feedback inspires us to enhance the clarity of arguments, particularly regarding the ladder-stacked structure.

However, I have serious doubts about several aspects of the work, and I believe that the following queries must be thoroughly answered before any recommendation can be made:

- 1) First, the Authors do not clearly describe how they created the samples that they call '20T-TaS₂'. They mention in the Methods section that they annealed the crystals after growth, but they do not actually attribute the transformation of 1T-TaS₂ into 20T-TaS₂ to this annealing process. At the very least, in order for the main results of the paper to be reproducible by other groups, the Authors should explain how they were able to synthesize 20T-TaS₂ crystals. Can the Authors demonstrate that the procedure is reproducible? How many of each kind of sample (1T-TaS₂ and 20T-TaS₂) were created for this work?

Author reply:

The synthesis of formed named 20T-TaS₂ crystals (renamed as LC-TaS₂) is quite normal. Initially, we obtained the 1T-TaS₂ crystals using the detailed procedure outlined in the Method section. Subsequently, these 1T-TaS₂ crystals were transferred to a vacuum chamber (better than 10⁻² mbar) and annealed at 150-200 °C for 3-4 hours. The LC-TaS₂ crystals are obtained after cooling to room temperature naturally. A similar post-treatment procedure has been adopted in our previous report, (*ACS Nano* **12**, 12619-12628 (2018)) where a higher annealing temperature triggered a surface structure phase transition. To improve reproducibility, we provide a temperature/time range due to potential uncertainties during thermal equilibrium caused by the misalignment between the thermocouple and mounted crystals.

The transformation of 1T-TaS₂ crystals into LC-TaS₂ crystals is achieved through this process. The quantity of LC-TaS₂ crystals collected is subjected to the crystals mounted to the heating plate and the uniformity of heating plate.

The STEM, Synchrotron-based XRD, and XPS results shown in this work were obtained from the same crystals and can be reproduced on crystals of other batches monitored by lab-based XRD (left panel in Fig. R1 on page R3 in this letter). STM, ARPES and transport measurements, however, were performed on crystals qualified by lab-based XRD. The consistency observed in the results indicates the reproducibility of our procedure.

Change to the manuscript:

We inserted the following sentences in Methods section on page 11.

“Briefly, the polycrystalline TaS₂ powder and iodine (as a transport agent) were sealed in quartz tubes under a vacuum. The quartz tubes were heated in a gradient furnace for 10 days. The 1T-TaS₂ crystals were obtained after quenching the quartz tubes in an ice-water mixture. The obtained 1T-TaS₂ crystals were transferred to a vacuum chamber (better than 10⁻² mbar) and annealed at 150-200 °C for 3-4 hours. The LC-TaS₂ crystals are obtained after cooling to room temperature naturally.”

2) This may be the most important query: The structure that is introduced here, namely the $20T$ structure, is inferred from only one HAADF-STEM image with a tiny field-of-view. But it is unclear how the Authors know that the other measurements (STM and ARPES) are also performed on a $20T$ -structured region of the sample. It seems necessary to assume that the $20T$ structure is uniform throughout the whole of some crystals, and totally absent in other crystals, and that the Authors can use some method (e.g. XRD) to sort these crystals from each other. The justification for these assumptions needs to be demonstrated. Can the Authors demonstrate that the $20T$ structure can be uniformly observed throughout multiple parts of the same sample?

Author reply:

As detailed in the former Author reply section, the LC-TaS₂ crystals, crucial to our investigation, originate from the transformation of $1T$ -TaS₂ crystals. The ladder-stacked structure is clearly illustrated in HAADF-STEM images.

We had quantified the quality of transformed LC-TaS₂ samples by using the lab-based XRD, as commented by Reviewer. As shown in left panel of Fig. R1, a series of diffraction peaks between 10° to 70° could be fully indexed to (00 l) planes of LC-TaS₂ crystals, verifying the high crystallinity of $1T$ - and LC-TaS₂ samples.

Fig R1. (left) XRD patterns for one $1T$ -TaS₂ and three LC-TaS₂ crystals, along with substrate signals. (right) Synchrotron-based XRD patterns for $1T$ -TaS₂ and LC-TaS₂ crystals shown in Fig. 1 of main text.

However, it is hard to verify the difference between $1T$ -TaS₂ and LC-TaS₂ crystals by using lab-based XRD. To address the crucial question regarding the homogeneity of the laddering stack structure in the transformed samples, we monitored the structural evolution using synchrotron-based XRD (shown as Fig.1c in the manuscript and represented in right panel of Fig. R1). The monochromatic synchrotron provides an enhanced resolution. This method allows us to verify and ensure the dominance of the laddering stack structure in the crystals under

investigation, and quantify the 0.03 Å interlayer distance difference between LC- and 1T-TaS₂ crystals. The residual 1T-stacking structure with a lower $q = 10.69 \text{ nm}^{-1}$ indicate a non-complete transition. It is not surprising that a complete transition to the metastable structural phase remains a great challenge as the local fluctuation favors the relative more stable 1T-phase. (*ACS Nano*, **6**, 7311–7317 (2012); *Nat. Commun.*, **8**, 486 (2017))

Importantly, the selection of crystals for subsequent STM and ARPES measurements is contingent upon their qualification through lab-XRD analysis.

The uniformity of the laddering stack structure within a single crystal is evidenced by the HAADF-STEM images captured from different sections of samples cut from the same crystal, as shown in Fig. R2. These STEM results confirm the prevalence of the laddering stack structure across multiple parts of the crystal.

Fig. R2. An additional cross-sectional HAADF-STEM image along the (100) direction for LC-TaS₂ crystals. The white horizontal and vertical solid lines indicate the in-plane and out-of-plane direction, respectively. The red solid lines highlight the atomic misalignment.

Change to the manuscript:

a) We insert the following sentence or words on Method section on page 11-12.

“All collected crystals were stored in an Ar-protected glove box before further characterization. Subsequently, the crystals were cut to pieces for electron microscopy, synchrotron-based XRD and PES characterization.

...

The ARPES measurements were performed at the BL13U beamline of NSRL. The single crystals, previously qualified by XRD, were cooled to 280 K, cleaved and measured *in-situ* in a UHV chamber with a base pressure of 1×10^{-10} mbar.

...

The STM/STS measurements were conducted in an Omicron LT STM system. The obtained 1T- and LC-TaS₂ single crystals, qualified by XRD, were cleaved and transferred into a UHV prep-chamber with a base pressure of 5×10^{-10} mbar.

...

The electrical transport measurements for freshly cleaved single crystals, cut from XRD-qualified samples, were carried out by using a physical property measurement system (PPMS, Quantum Design) with a standard six-probe method.”

b) We also inserted the following sentence and references on page 5.

“A shoulder with a lower $q = 10.69 \text{ nm}^{-1}$ originates from residual *bb* stacking configurations. It is not surprising that a complete transition to the metastable laddering stack structural phase remains challenging, given that local fluctuation favors the relative more stable *bb* stacking 1T-phase.”^{35, 36}

³⁵ Eda, G. *et al.* Coherent Atomic and Electronic Heterostructures of Single-Layer MoS₂. *ACS Nano* **6**, 7311-7317 (2012).

³⁶ Yin, X. *et al.* Tunable inverted gap in monolayer quasi-metallic MoS₂ induced by strong charge-lattice coupling. *Nat. Commun.* **8**, 486 (2017).”

- 3) Can the Authors clearly demonstrate that the apparent shear seen in the HAADF-STEM image (Fig. S1 for example) really corresponds to a ‘ladder stacking’ in the sample, and not a shear distortion due to the properties or calibration of the microscope?

Author reply:

The HAADF-STEM images for 1T-TaS₂ and LC-TaS₂ samples were uniformly captured by using the same instrument and during the same timeframe.

The self-consistency of sliding along (100) (110) and ($\bar{1}\bar{2}0$) directions for LC-TaS₂ and well alignment along (100) and (110) directions for 1T-TaS₂

demonstrates that observed shear is not an artifact resulting from the properties or calibration of the microscope. Please refer to reply to the following comment for further details.

- 4) The stacking displacement between two layers is a 3D vector, which has two in-plane components. To establish both of these in-plane components, it is necessary to take at least two HAADF-STEM images of the same sample from different directions, e.g. one along the (100) direction and one along the (010) direction. Can the Authors do this? If not, how do we know what the in-plane stacking displacement is?

Also, how do the Authors know that the sample that they call 1*T*-TaS₂ does *not* have a shear distortion if they have only viewed it from one direction?

Author reply:

We appreciate the Reviewer to point out this important question.

To clarify the in-plane stacking displacement of LC-TaS₂ crystals, the HAADF-STEM images were collected along the (100), (110), ($\overline{120}$) directions. The corresponding images along the ($\overline{120}$) and (110) are presented in Fig. 1i and Supplementary Fig.S3 (both attached below), respectively. It is concluded that the interlayer sliding occurs along the ($\overline{120}$) direction within the *ab*-plane, as schematically illustrated in Fig. 1h. The Ta-atoms in the upper layer are expected to align with the Ta-atoms in the lower layer after a 40-layer periodicity. And a 40-layer periodicity is confirmed from image along the (110) direction (Supplementary Fig.S3). Consequently, we correct the periodicity from ‘20-layer’ guided by image along (100) direction to ‘40-layer’ as evidenced by sliding along ($\overline{120}$).

To address the concern regarding shear distortion in 1*T*-TaS₂, HAADF-STEM images were collected along the (100) and (110) directions. The images, shown in Supplementary Fig. S1 (also attached below), reveal a well-aligned on-top Ta-atoms stacking configuration along the *c*-direction. This, coupled with single XRD (001) peak shown in Fig. 1c, rules out the possibility of shear distortion in 1*T*-TaS₂, which is consistent with literature reports, (*Proc. Natl. Acad. Sci. U.S.A.* **113**, 11420-11424 (2016)) confirming the higher quality of the obtained 1*T*-TaS₂ crystals.

Change to the manuscript:

- a) We inserted/changed the following sentence on page 4 in the main text and added the corresponding figure in Fig. 1.

“The laddering stack structure is verified using aberration-corrected high-angle annular dark-field scanning transmission electron microscopy (HAADF-STEM), as shown in Fig. 1k. This unique structure features a routine sandwiched unit, whereas a periodically misaligned interlayer stacking arrangement along the *c*-direction (Fig. 1j) distinguishes it from the well-established vertical alignment of

Ta-atoms between adjacent layers in $1T$ -TaS₂ (Figs. 1a and 1b, and Supplementary Fig. S1).³³ The interlayer sliding occurs along the $(\bar{1}20)$ direction within the ab -plane guided by the red arrows in Fig. 1h and 1g, as determined from corresponding HAADF image (Fig. 1i). Viewing along the (100) direction, a 20-layer periodicity is discerned from a large-scale HAADF image (see Supplementary Fig. S2). Considering the $(\bar{1}20)$ direction interlayer sliding, the Ta-atoms in the upper layer are expected to align with the Ta-atoms in the lower layer after a 40-layer periodicity, corresponding to a translational shift of ~ 0.015 nm per layer. The 40-layer periodicity is confirmed by HAADF image along (110) direction (Supplementary Fig. S3). Here, we name the crystals with octahedral coordinated TaS₂ units stacking in laddering configuration as LC-TaS₂.

Figure 1. Variation in the atomic structure and insulating nature. **a** Schematic side view of $1T$ -TaS₂ crystals, and **b** corresponding (100) cross-sectional HAADF-STEM image. **c** Synchrotron-based XRD spectra around the (001) diffraction peak collected at room temperature. **f** The out-of-plane superlattice and reflections for a $1T$ -TaS₂ crystal collected at 120 K. The peak splitting is attributed to the Cu $K_{\alpha 1}$ and $K_{\alpha 2}$ X-ray split. Illustration of **d** the David-stars T_A stacking and **g** the T_A^S stacking associated with **j** the laddering stack structure in LC-TaS₂ crystals verified by HAADF-STEM images collected along **k** (100) and **i** $(\bar{1}20)$ directions. The white vertical lines in panels **b**, **k** and **i** indicate the c -direction. The red solid line in panel **k** highlights the atomic misalignment. **h** Schematic top view of monolayer octahedral TaS₂. The red and purple arrows indicate the $(\bar{1}20)$ and (110) directions, respectively. **e** Schematic energy band diagrams depicting a band-insulator and a Mott-insulator.”

- b) We displayed HAADF-STEM images along the (100) and (110) directions for 1T-TaS₂, image along (110) direction for LC-TaS₂ in supporting information.

Fig. S1. Vertical aligned atomic stack structure. The cross-sectional HAADF-STEM images for 1T-TaS₂ crystals collected along the (a) (100) and (b) (110) directions. The white horizontal and vertical solid lines indicate the in-plane and out-of-plane direction, respectively.

Fig. S3. Laddering stack structure. The cross-sectional HAADF-STEM images for LC-TaS₂ crystals collected along the (110) direction.”

- 5) The Authors note that the STM conductance spectra shown in Figure 2(c) likely corresponds to the so-called T_A -stacked and T_C -stacked terminations of the layer stacking pattern of the CDW [described in Ref. 20]. But the figure strongly suggests that the difference between the two conductance curves can be attributed to the difference between the $1T$ and $20T$ structures. This is misleading. If the Authors want to make a comparison between STM measurements on the $1T$ and $20T$ samples, I think they should find the directly comparable surfaces of each case, i.e. the T_A surface of $1T$ and $20T$ samples for comparison (and likewise, the T_C surface of $1T$ and $20T$ samples). I think this would allow a proper comparison to the DFT and ARPES results.

Author reply:

The surface sensitivity of STM technique poses at challenging in provide information about the second layer, hindering the direct identification of the David-stars stacking configuration. As an indirect observation, we attempted to image the step region of LC-TaS₂ surface using STM. As shown in the

Supplementary Fig. S6a (also attached below), both T_A^S -stacking (guided by black dots) and T_C^S -stacking (guided by blue and black dots) configurations are proposed, assuming the lower layer maintains the uniform CDW domain patterns. However, this is not the case as three domains guided by different color dots are observed. We thus cannot conclusively demonstrate the David-star stacking configuration for which conductance spectra were obtained.

To ensure uniformity, we conducted a series of STS curves at various locations on the LC-TaS₂ surface, as shown in Supplementary Figs. S6c and S6d. These experiments showed that the smaller Mott-gap for LC-TaS₂ is insensitive to terminations. We also conducted a series of STS curves on 1T-TaS₂ surface, which align with T_A stacking reported previously. (*Nat. Commun.* **11**, 2020, 2477; *Phys. Rev. B* **105**, 035109 (2022)) The uniformity is commented by Reviewer 3 [comment 4)]. Please refer to Author reply for additional details on page R40.

In addition, the two STM conductance curves in Fig. 2(c) were obtained from two separated 1T-TaS₂ and LC-TaS₂ crystals, rather than across different terminations of the same crystal. But these curves do exhibit similarities to the results obtained from T_A -stacked and T_C -stacked terminations of the same 1T-TaS₂ crystal. (*Nat. Commun.* **11**, 2477 (2020); *Phys. Rev. B* **105**, 035109 (2022)) In those reports, T_A -stacking features CDW dimerization with a larger band gap, whereas the split T_C -stacking features a relative smaller Mott-gap. The similarities on narrow energy gap between T_C -stacked terminations and sliding structure imply their common origin of interlayer decoupling.

Change to the manuscript:

a) We changed the following sentences on page 6 from

“Similar STS spectra, particularly those with a narrow energy gap, have also been theoretically simulated and experimentally observed on the cleaved 1T-TaS₂ surface with specific David-stars T_C stacking.^{13, 20, 21}”

To

“The STS spectra with a narrow energy gap have also been theoretically stimulated and experimentally observed on the cleaved 1T-TaS₂ surface with specific David-stars T_C stacking,^{13,20,21,42} suggesting their common origin of interlayer decoupling.”

b) We displayed the STM image at the step region of LC-TaS₂ surface in the supporting information.

“**Fig. S6.** (a) The STM topography of the step area for LC-TaS₂. (b) The STM image with lower contrast to highlight the CDW domain pattern in upper layer. The color dots guide the David-stars centers. The inset shows a height profile across the step. Assuming the lower layer maintains uniform CDW domain patterns, configurations of both T_A^S-stacking (guided by black dots) and T_C^S-stacking (guided by blue and black dots) configurations are proposed in panel (a). A series of dI/dV spectra with (c)(e) linear intensity scale and (d)(f) logarithmic intensity scale measured at different surface locations of LC-TaS₂ and 1T-TaS₂.”

-
- 6) Somewhat related to the point above: Is it possible that the difference between the ARPES results shown in Figs. 3(a-c) and Figs. 3(d-f), and also between Figs. 3(g) and 3(h), could correspond to observations of the T_A -stacked and T_C -stacked surface layers, and not to the observation of the $1T$ and $20T$ structures?

Author reply:

In consideration of the mentioned point, it is conceivable that the observed dispersive bands and flat bands from ARPES spectra could be attributed to the surface layers exhibiting T_A -stacked and T_C -stacked configurations in $1T$ -TaS₂ rather than the $1T$ and ladder-like stack structures. However, this scenario can be ruled out because:

(i) From a probability viewpoint, both the T_A -stacked and T_C -stacked surface layer can be obtained after cleavage. However, assuming an equal likelihood, the exposed domains of T_A -stacked and T_C -stacked configurations are expected to be the same. In this regard, ARPES spectra would receive a similar contribution from dispersive bands and flat bands, rather than primarily from a flat band as shown in Figs. 3e and 3f.

(ii) Ultimately, even if a uniform T_C -stacked surface was fortuitously obtained on surface of samples, the flat band feature would remain elusive. This is attributed to the detected depth of ARPES spectra, which is 2-3 nm for electrons with 18 eV kinetic energy. (*Surf. Interface Anal.*, **1**, 2 (1979)) This depth corresponds to at least three layers (~0.6 nm/layer), (*Nat. Commun.*, **11**, 4215 (2020)) comprising one T_C -stacked layer and one T_A -stacked bilayer. As a result, ARPES spectra would also receive two set contribution of dispersive bands and flat bands.

To sum up, the distinct flat bands can solely originate from the ladder-like stack structures with decoupled interlayer interaction. And the dispersive bands for $1T$ -TaS₂ structures receive a dominant contribution from T_A -stacking and a minor one from T_C -stacking. This conclusion aligns with our finding on the dualistic insulator states in $1T$ -TaS₂ crystals.

Change to the manuscript:

We insert the following sentences and references on page 9.

“The ARPES spectra, with a detected depth of 2-3 nm for electrons with 18 eV kinetic energy, extend across three layers,⁴⁷ providing multilayer information rather than characteristics of top surface layer.²² In LC-TaS₂, the 2D flat bands close to E_F along both the Γ -A/ k_z and Γ -M/ $k_{||}$ directions in the ARPES measurements and the wide energy gap determined from the STS spectra, in agreement with DFT calculations confirms its 2D Mott-insulator nature.

⁴⁷ Seah, M. P. & Dench, W. A. Quantitative electron spectroscopy of surfaces: A standard data base for electron inelastic mean free paths in solids. *Surf. Interface Anal.* **1**, 2-11 (1979).

22 Wang, Y. D. *et al.* Band insulator to Mott insulator transition in 1T-TaS₂. *Nat. Commun.* **11**, 4215 (2020).”

- 7) The Authors perform DFT calculations to anticipate the electronic properties of the 20T structure. They include an in-plane displacement of 0.39 Angstroms or more and observe that as the displacement increases, the gap gradually closes. (They show that the gap remains open if the displacement is only 0.39 Angstroms.) The in-plane displacement in the putative 20T-TaS₂ sample would be only about 0.17 Angstroms, and it seems clear from the trend shown in Fig. S5 that there will be a gap, *without* the Hubbard U , for a displacement of 0.17 Angstroms.

Author reply:

We address both comment 7) and 10) collectively to avoid redundancy, as they share a comment related to larger in-plane displacement parameter in our DFT calculations deviated from experimental value.

- 10) I fail to understand why the Authors did not simply perform the DFT calculation with an in-plane shift of about 0.17 Angstroms, directly corresponding to the 20T-TaS₂ structure. I understand that the Authors can refer to multiple values of in-plane displacement to emphasize that it may be a control parameter to adjust electronic structure in what they call ‘slidetrionics’, but at the very minimum it is necessary to present a DFT treatment that corresponds exactly to the sample that is observed in experiment, to establish that the basic idea is valid.

Author reply:

We appreciate the Reviewer’s valuable suggestion. Performing DFT calculation precisely matching the 20-layer unit cell containing 780 Ta- and S-atoms (39 atoms in a CDW star-of-David superstructure for one layer) by using the DFT+U method is highly challenging, especially for precise evaluation of small energy-gap systems. Specifically, in the revised manuscript, the corrected 40-layer periodicity involves 1560 Ta- and S-atoms. Consequently, due to computational constraints and to align with the methodology adopted in most prior studies, we have adopted a simplified 4-layer supercell structure containing 156 Ta- and S-atoms with a complete period of T_AT_C stacking.

Our simplified approach enables us to identify critical values of in-plane displacement that could potentially close the energy gap. In the case of the 4-layer supercell structure, a $1/6a$ in-plane displacement, which is lower than $\sqrt{3}/4a$, can close the gap (Supplementary Fig. S10 and also attached below for convenience). We understand that this simplified model has inherent limitations for simulating the experimental observed sliding effect. As pointed out by the Reviewer and corroborated by Figure S5, the gap remains open when the in-plane displacement is 0.17 Å (corrected to be ~ 0.15 Å for sliding along $(\bar{1}20)$ direction). We attribute this mismatch to the reduction in the number of the supercell layer.

Fig. S10. DFT calculated band structures of 4-layer supercell structures. Band structure for $T_A^S T_C^S$ stacking with different sliding distances δa of 0.39 Å (1/8a), 0.53 Å (1/6a), 0.76 Å (1/4a), and 1.06 Å (1/3a) (a-d) without Hubbard U , and (e-h) with Hubbard U , respectively. Without considering Hubbard U , the energy gap gradually decreases as the sliding distance increases and closes for δa exceeding 0.53 Å (1/6a). The introduction of U leads to the reopening of energy gap for all δa values.

To clarify the coordination effect between supercell layer numbers and sliding distance in representing experimental electronic structures, we expanded our calculations to 8-layer supercell structures without Hubbard U , representing our current calculation limits. Supplementary Fig S11 illustrates the corresponding band structures for multiple values of in-plane displacement. Figure S11e summarized the evolution of calculated energy gap as a function of in-plane displacement for both 4- and 8-layer supercell. Notably, a 0.32 Å (1/10a) value is found to close the gap for a larger 8-layer supercell structure, which is smaller than 0.53 Å (1/6a) required for the 4-layer supercell structure. Figure S11f represents the evolution of in-plane displacement $\sqrt{3}/na$ for n -layer supercell structures. The in-plane displacements that could close the gap, as extracted from calculations, are consistently smaller than $\sqrt{3}/na$. We thus conclude that $\leq \sqrt{3}/na$ in-plane displacement can close the gap for an n -layer supercell structure. Therefore, it is reasonable to propose that $\leq \sqrt{3}/40a$ in-plane displacement can close the gap of the 40 layers supercell structures, as observed experimentally.

Our results not only contribute to understanding the evolution of ‘slidronics’ by controlling in-plane displacement, but also provide an effective dimensional simplification strategy (enlarging in-plane displacement) to represent the experimental results influenced by subtle sliding distance.

Change to the manuscript:

a) We inserted the following sentence in Method section on page 13.

“Simulating the experimentally observed 40-layer supercell structure, which contains 1560 Ta- and S-atoms, poses challenges. As a result, a simplified structure with a minimum of 4 layers or one $T_A^S T_C^S$ stacking period was adopted. To compensate for the effects of this structural simplification, a larger sliding distance δa of 0.053 nm (1/6a) was used. The validity of this approach was verified by varying δa from 0.039 nm (1/8a) to 0.106 nm (1/3a). Supplementary Fig. S5 demonstrates that the band structures calculated with $\delta a \geq 0.053$ nm (1/6a) accurately reproduce results from the 4 layers supercell structure. To clarify the coordination effect between supercell layer numbers and sliding distance in representing experimental observations, extended calculations of 8-layer supercell structures were performed. Supplementary Fig S11 illustrates the corresponding band structures for various δa . Notably, a 0.032 nm (1/10a) value is identified to close the gap for a larger 8-layer supercell structure, which is smaller than 0.053 nm (1/6a) for the 4-layer supercell structure (Fig.S11e). The observations lead to the conclusion that a sliding distance $\leq \sqrt{3}/na$ can effectively close the gap for an n -layer supercell structure (Fig. S11f). Therefore, it is reasonable to propose that $\sqrt{3}/40a$ sliding can close the gap in a 40-layer supercell structure, as observed experimentally.”

b) We displayed the DFT results of 8-layer supercell structures in Supporting information

Fig. S11. DFT calculated band structures of 8-layer supercell structures. Band structures for $T_A^S T_C^S$ stacking with different sliding distances δa of (a) 0.26Å (1/12a), (b) 0.32Å (1/10a), (c) 0.39Å (1/8a), and (d) 0.53Å (1/6a), respectively. Without considering Hubbard U , the energy gap gradually decreases as the sliding

distance increases and closes for δa exceeding 0.32 \AA ($1/10a$). (e) The evolution of calculated energy gap as a function of sliding distance δa for the 4- and 8-layer supercell. (f) The evolution of in-plane displacement $\sqrt{3}/na$ for n -layer supercell structures, along with minimum sliding distance value that close the gap. The gap value in panel e is taken from the difference of the valence band maximum (VBM) and conduction band minimum (CBM). The negative gap value indicates a metal state.”

- 8) In the DFT calculations modelling the $1T$ -TaS₂ case [e.g. Fig. 4(d)], why do the Authors only show GGA results and not GGA+U? The on-site Coulomb repulsion must still exist in the layer-dimerized band insulating $1T$ -TaS₂, even though it is not the primary reason for the band gap to appear.

Author reply:

We address both comment 8) and 9) collectively to avoid redundancy, as both center around concerns related to the DFT calculations of $1T$ -TaS₂ without Hubbard U term.

- 9) Related to the point above, the Authors present a comparison between GGA band structure for the $1T$ structure, and GGA+U band structure for the $20T$ structure [Figs. 4(a) and 4(h)], thereby making a comparison when two variables are changed simultaneously. This is confusing. To me it seems that the GGA+U results for each case should be compared.

Author reply:

We acknowledge the Reviewer’s suggestion that on-site Coulomb repulsion on David-stars within ab plane still exists in the layer-dimerized band insulating $1T$ -TaS₂, and have conducted the DFT calculations incorporating GGA+U, as suggested. The results are now included in Fig. 4 of the revised manuscript and are also attached below for your convenience.

It is evident from Fig. 4c and 4e that the gap calculated with GGA+U method is slightly larger than that obtained with GGA. Notably, the unfolded band structures below Fermi level exhibit minimal changes. This observation underscores our expectation that on-site Coulomb repulsion is not the primary reason for the band gap in $1T$ -TaS₂. In order to emphasize this point and facilitate a more comprehensive comparison of the DFT and experimental results, we have incorporated the corresponding figure into Fig. 4. We trust that this addition enhances the clarity and comparison of our results, considering one variable.

Change to the manuscript:

- a) We inserted the following sentences on page 8.

“The band structure for 4 layers $1T$ -TaS₂ (Fig. 4d) with David-stars $T_A T_C$ stacking (Fig. 4a) is in good agreement with previous reports.^{14,15,46} However, for $T_A^S T_C^S$ stacking (Fig. 4b) with a translational shift of 0.053 nm , the calculated band

structures exhibit metallic characteristics with a conduction band crossing the E_F at the L point (Fig. 4g). To address this discrepancy, the Hubbard U term is introduced in calculations, which opens a gap (Fig. 4h), confirming its Mott-insulator nature. For comparison purposes, the GGA+ U method is also employed to calculate the band structure, as shown in Fig. 4e. As expected, the limited increase in the energy gap after the introduction of the Hubbard U term indicates that on-site Coulomb repulsion is not the primary factor affecting the energy gap for $T_A T_C$ stacking (Fig. 4c). In addition, interlayer distance contraction has negligible influence on the low-energy electronic structure beyond the reduced gap (Supplementary Fig. S8), confirming the dominant roles that ladder-like stack played in determining the interlayer coupling. The validity of simplifying the system to only 4 layers and employing a larger translational shift is discussed in the Methods part.”

b) We updated the Fig. 4 and corresponding figure caption from

Fig. 4. DFT calculated band structures. Schematic diagrams illustrating (a) the $T_A T_C$ and (e) $T_A^S T_C^S$ David-stars stacking sequence. Band structures near the E_F for (b) $T_A T_C$ stacking calculated using the GGA method, and for (f) $T_A^S T_C^S$ stacking using GGA method and (g) with additional Hubbard U term (GGA+ U). (c) DOS for the $T_A T_C$ and $T_A^S T_C^S$ stacking. Calculated unfolded band structure for (d) $T_A T_C$ (GGA) and (h) $T_A^S T_C^S$ (GGA+ U) stacking.”

To

“

Figure 4. DFT calculated band structures. Schematic diagrams illustrating **a** the $T_A T_C$ and **b** $T_A^S T_C^S$ David-stars stacking sequence. **c** DOS for the $T_A T_C$ and $T_A^S T_C^S$ stacking. Band structures near the E_F for $T_A T_C$ stacking calculated using the **d** GGA and **e** GGA+U method. Band structures for $T_A^S T_C^S$ stacking using **g** GGA and **h** GGA+U method. Calculated unfolded band structure for **f** $T_A T_C$ and **i** $T_A^S T_C^S$ stacking using GGA+U method.”

- 11) A minor query: The Authors interpret the XRD as indicating that the inter-layer spacing is reduced for the $20T$ structure, but they interpret the XPS as indicating that the $20T$ structure has relatively decoupled layers (line 119). Are these interpretations consistent with each other?

Author reply:

XRD indicates a reduced out-of-plane inter-layer spacing for LC-TaS₂. Without considering the in-plane displacement, we could image an enhanced inter-layer coupling due to the larger spatial overlap between the wavefunctions of adjacent layers associated with contraction. However, this assumption is subjected to the

experimental evidence. For instance, Zhang group found that the inter-layer spacing reduce from 5.928 Å in C-CDW state to 5.902 Å in intermediated state evidenced by temperature dependent XRD is associated with an interlayer decoupling revealed by ARPES, featuring band-to-Mott transition at narrow temperature window (*Nat. Commun.*, **11**, 4215 (2020)). Although the detailed structural mechanism of Mott transition is unknown, interlayer decoupling is associated with the out-of-plane contraction triggered by heating.

On the other hand, XPS or photoemission spectroscopy (PES) spectra indicate charge redistribution of Ta-atoms within each in-plane David-stars. The enhanced electron localization points towards decoupled inter-layer interaction.

Taking the in-plane displacement into account, the apparent contradiction is well interpreted. The in-plane displacement acts to reduce the spatial overlap between the wavefunction, dominating over the out-of-plane contraction effect and, in turn, decoupling the inter-layer interaction.

To rule out the roles that interlayer distance contraction played in determining the electronic structure, the DFT calculations were performed. This issue is commented by Reviewer #2 [comment 6)]. To avoid redundancy, please refer to Author reply and Change to the manuscript sections on page R33-34 for details.

Reviewer #2 (Remarks to the Author):

Wang et al. report on a new structural modification that they found in the well studied layered van der Waals material $1T$ -TaS₂. They show that annealing a $1T$ -TaS₂ single crystal followed by cooling to room temperature creates a structure in which the layers are slightly shifted with respect to each other yielding a laddering stack structure which the authors call $20T$ -TaS₂ as the period of this structure along the c -axis is 20 layers. The authors then employ a comprehensive combination of (angle resolved) photo-emission spectroscopy (AR)PES, scanning tunneling spectroscopy (STS), x-ray diffraction (XRD) and density functional theory (DFT) to study the electronic structure of $20T$ -TaS₂ in comparison to $1T$ -TaS₂. The two main conclusions of this study are, firstly, that the well-known star-of-David (SOD) charge density wave (CDW) structure in $20T$ -TaS₂ is essentially the same as in regular $1T$ -TaS₂. Secondly, they conclude that due to the interlayer shifts in $20T$ -TaS₂ the interlayer hybridization between SODs in adjacent layers is effectively reduced which in turn gives rise to increases electron-electron correlations rendering $20T$ -TaS₂ as being a Mott-insulator.

The manuscript is well written and accessible to non-expert readers. While I find the discovery of the laddering stack structure $20T$ -TaS₂ certainly very interesting, I have major concerns about the robustness of the drawn conclusions as explained in detail below:

Author reply:

We appreciate the thorough review provided by Reviewer and the positive comments on the interesting discovery of the laddering stack structure.

We are grateful for the acknowledgment of the clarity of our writing and our efforts to make the content accessible to non-specialists.

The succinct conclusions precisely capture the essence of our study. The constructive feedback inspires us to robustly support the drawn conclusions, particularly regarding the CDW stacking.

- 1) My first point concerns the robustness of the assumption that the TaTc CDW stacking is indeed preserved in $20T$ -TaS₂. The STS and PES data nicely indicate that the SOD structure of the commensurate CDW is preserved at the surface of $20T$ -TaS₂. I think this is a rather robust result but maybe not utterly surprising as the surface layer is less effected by the sliding distortion than a layer deep in the bulk. However, it is well-known that the stacking order of the CDW itself very strongly effects the electronic structure. The authors assume that the alternating TaTc stacking is preserved and not effected by the sliding distortion. As far as I can see there is no experimental data to backup this assumption. It would be very interesting for instance to see the XRD superlattice reflections and in particular their peak-profiles in the direction perpendicular to the layers which directly reveals the CDW stacking. I am wondering if the GIXRD data perhaps already contain this information.

Author reply:

The $T_A T_C$ David-stars (SOD) stacking configuration is widely accepted in $1T$ -TaS₂ (cf. Fig. 4a and also shown below). T_A represents the ordered array of David-stars, forming the bilayer, whereas the T_C denotes an approximation to the disordered stacking of bilayers, as the disordered stacking David-stars does not obey translational symmetry along the c -axis.

Fig. 4 Schematic diagrams illustrating (a) the $T_A T_C$ and (b) $T_A^S T_C^S$ David-stars stacking sequence.

The synchrotron-based XRD data presented in the manuscript were performed at room temperature NC-CDW without superlattice formation. To approach the David-stars stacking configuration in LC-TaS₂, lower temperature XRD along the (00 l) direction was performed on $1T$ - and LC-TaS₂ crystals. As shown in Fig. 1f collected at 120 K (attached below), the superlattice reflections at $\sim 37.9^\circ$ and $\sim 54.1^\circ$ in $1T$ -TaS₂ crystals due to Davis-stars dimerization associated with $T_A T_C$ stacking (Fig. 4a) disappear in LC-TaS₂, implying the absence of ordered array of David-stars or collapse of CDW states in LC-TaS₂ crystals. The persistence of David-stars evidenced by STM image (Fig. 2e) confirms the absence of ordered array. This is expected as the atomic translational sliding breaks the CDW translational symmetry along the c -direction, resulting disordered stacking David-stars.

The corresponding $T_A^S T_C^S$ stacking configuration, associated with atomic layer sliding, representing one of configuration without ordered array of David-stars as illustrated in Supplementary Fig. S9 (attached below), was employed for electronic structures calculation of LC-TaS₂ crystals. This choice aimed to underscore the role that atomic layer sliding played in altering the electronic structures. The resulting changed electronic structures support that the stacking order of the CDW itself very strongly affects the electronic structure. (*Nat. Phys.* **11**, 328-331 (2015)) Consequently, we retained the schematic configuration in Fig. 1g unchanged, and left the precise disordered CDW stacking configurations an open question.

Change to the manuscript:

- a) We inserted the following sentences on page 5, and displayed the low temperature XRD superlattice reflections collected at 120K in Fig 1f and (00 l) pattern in Supplementary Fig. S4.

“The interlayer spacing d is determined by synchrotron-based X-ray diffraction (XRD) spectroscopy at room temperature (Fig. 1c). The scattering vector q for LC-TaS₂ is determined to be 10.75 nm⁻¹, corresponding to a $d = 0.584$ nm (calculated using $d = 2\pi/q$). This d value is slightly smaller than 0.587 nm ($q = 10.70$ nm⁻¹) for 1T-TaS₂,³⁴ indicating a slight out-of-plane contraction. A shoulder with a lower $q = 10.69$ nm⁻¹ originates from residual bb stacking configurations. It is not surprising that a complete transition to the metastable laddering stack structural phase remains challenging, given that local fluctuation favors the relative more stable bb stacking 1T-phase.^{35,36}

To understand the influence of laddering stack structure on interlayer dimerization, the XRD measurements were performed using Cu K_{α} . Fig.1f displays the superlattice reflections collected at 120 K, and the (00 l) indexed diffraction peaks are presented in Supplementary Fig. S4. In the 1T-TaS₂ crystal, the interlayer dimerization associated with David-stars ordered array results in half-integer (0,0,5/2) and (0,0,7/2) reflections at $\sim 37.9^{\circ}$ and $\sim 54.1^{\circ}$, consistent with previous findings.²² The absence of these superlattice reflections in the LC-TaS₂ crystal implies the collapse of interlayer dimerization, either caused by collapse of CDW states or the absence of David-stars ordered array.

Figure 1. Variation in the atomic structure and insulating nature. c Synchrotron-based XRD spectra around the (001) diffraction peak collected at room temperature. f The out-of-plane superlattice reflections for a 1T-TaS₂ crystal collected at 120 K. The peak splitting is attributed to the Cu $K_{\alpha 1}$ and $K_{\alpha 2}$ X-ray split.

Fig. S4. Low temperature XRD characterization. XRD patterns for 1T-TaS₂ and LC-TaS₂ crystals measured at 120 K and 300 K, respectively.”

b) We inserted the following sentence on page 6.

“Remarkably, the long-range CDW order remains intact at the surface of LC-TaS₂ crystals. The persistence of David-stars, along with absence of out-of-plane superlattice reflections (Fig. 1f), implies the absence of out-of-plane ordered array of David-stars.”

c) We inserted the following sentence on page 7.

“Furthermore, we performed DFT calculations to gain a comprehensive understanding of electronic structures in the above STS and ARPES experiments. A 4 layers supercell with an ordered David-stars T_AT_C stacking (Fig. 4a), identified by XRD superlattice reflections in 1T-TaS₂ (Fig. 1k), was adopted. This configuration has been extensively employed in previous DFT calculations.^{13,14,20} The absence of out-of-plane superlattice reflections in LC-TaS₂ crystals indicates a disordered David-stars stacking configuration. Aiming to emphasize the influence of atomic layer sliding on the alteration of the electronic structures, T_A^ST_C^S stacking (Fig. 4b), integrating David-stars T_AT_C stacking with a translational shift representing disordered David-stars stacking due to broken of translational symmetry along the *c*-direction, was adopted. The band structure for 4 layers 1T-TaS₂ (Fig. 4d) with David-stars T_AT_C stacking is in good agreement with previous reports.^{14,15,46} However, for T_A^ST_C^S stacking with a translational shift of 0.053 nm, the calculated band structures exhibit metallic characteristics with a conduction band crossing the *E_F* at the *L* point (Fig. 4g).”

d) We inserted the XRD experimental details on page 11.

“XRD Measurement.

The synchrotron-based grazing incidence X-ray diffraction (GIXRD) measurements were performed at the BL14B1 beamline of the Shanghai Synchrotron Radiation Facility (SSRF) using X-rays with a wavelength of $\lambda = 1.24$ Å and grazing incident angles of 1.0° . The corresponding spectra collected at room temperature are presented in scattering vector q coordinates by using the equation $q = 4\pi\sin\theta/\lambda$, where θ is half of the diffraction angle. The q has been calibrated by measuring the synchrotron-based XRD of a lanthanum hexaboride reference sample.

Lab-based XRD characterizations were performed on a Bruker D8 diffractometer using Cu K_α radiation to assess the quality of the obtained crystals. XRD data were also collected on 1T- and LC-TaS₂ crystals after cooling down to 120 K.”

e) We inserted the following sentence on page 9. Fig S9 is also displayed in Supporting Information.

“In LC-TaS₂, the 2D flat bands close to E_F along both the Γ -A/ k_z and Γ -M/ $k_{||}$ directions in the ARPES measurements and the wide energy gap determined from the STS spectra, in agreement with DFT calculations confirms its 2D Mott-insulator nature. On the other hand, significant band dispersion in 1T-TaS₂ implies its classification as a 3D band insulator.²² It is noteworthy that, although DFT calculations based on T_A^S T_C^S stacking generally capture the reduced band dispersion character in LC-TaS₂, deviations in actual band dispersion values are present. This discrepancy may arise from alterations in the David-stars stacking sequence induced by atomic layer sliding as illustrated in Supplementary Fig. S9, warranting further investigation.

Fig. S9. Potential configurations of disordered David-stars stacking in 4-layer supercells. (a) $T_A^S T_C^S$ and (b,c) its random stacking counterpart, (d) $T_A^S T_B^S$, (e) $T_A^S T_B^S T_C^S$, (f) T_A^S , (g) T_B^S , (h) T_C^S , and (i) $T_B^S T_C^S$. Note that (g-h) do not contain T_A^S stacking observed in STM image (Fig. S6).”

- 2) My next point concerns the DFT results: It is not clear to me how the band structure shown in Fig 4 (b) and the unfolded band structure in Fig 4 (d) are related? In Fig 4 (b) there is a hole like band at the Gamma point which is completely absent in the unfolded band structure. Likewise the band between Gamma and A looks very different in Fig 4(b) and (d). The same issues occur for Fig 4 (f) and (h). Are these band structures really calculated using the same crystal structure as the text and caption suggests? While the unfolded band structures in Fig 4 (d) and (h) are more or less consistent with the literature, I am not aware that this hole like band at the Gamma point which disperses to -0.1 eV at the M point has been reported before for the TaTc stacking with periodic boundary conditions. I think the

authors need to revisit their calculations or better explain to which crystal structures they belong and how they relate to previously published electronic structures.

Author reply:

We appreciate the Reviewer's insight comments and would like to provide additional clarification regarding the supercell band and unfolded band.

The observed hole-like band observed around Γ point, as mentioned by Reviewer, arises due to band folding. For the sake of clarity, we present the calculated band structure of the CDW superstructure using 1-layer, 2-layer, and 4-layer supercells. Fig. R3a illustrates the corresponding Brillouin zones (BZ). Doubling the number of layers causes the reciprocal lattice parameter along the k_z direction to halve, resulting in band structures folding into the new BZ. Specifically, band structures in the $k_z^{1L}=\pi/4$ plane are shown in the $k_z^{2L}=\pi/2$ plane with a doubling of layers. Similarly, the band structures in the $k_z^{1L}=\pi/2$ plane fold into the $k_z^{2L}=0$ plane. This folding relation continues, and band structures in the $k_z^{2L}=\pi/2$ plane fold into $k_z^{4L}=0$ plane when the number of layers doubles again.

Fig. R3. (a) Brillouin zones for supercells with different numbers of layers, indicated by different colors. When the number of layers doubled, the reciprocal lattice parameter in the k_z direction will be halved. (b-d) Band structures for 1-layer, 2-layer, and 4-layer supercells. The hole pocket near Fermi energy pointed by the red arrow will be folded to $k_z^{4L}=0$ plane in 4-layer supercell (Fig. (d)). (e) Band structures for $T_A T_C$ stacking.

(f) Figure extracted from ref (*Mater. Res. Express* **10**, 046302 (2023)).

For the 1-layer structure (Fig. R3b), the calculated band structures shows a flat band along the Γ - M - K direction below the Fermi energy, consistent with previous reports. (*Phys. Rev. B* **96**, 125138 (2017)) In the right panel of Fig. R3b, the corresponding band structures along the k -path in the $k_z^{1L}=\pi/4$ plane exhibit a hole pocket around the A_{2L} point (guided by a red arrow). The doubling of layers results in the appearance of the hole pocket in the $k_z^{2L}=\pi/2$ plane, as depicted in Fig. R3c. Further doubling of layers to the 4-layer structure with $\mathbf{T}_A\mathbf{T}_A$ stacking leads to the folding of the hole pocket from the $k_z^{2L}=\pi/2$ plane to the $k_z^{4L}=0$ plane (Fig. R3d). The introduction of $\mathbf{T}_A\mathbf{T}_C$ stacking further modifies the band dispersion and opens a gap (Fig. R3e). The hole-like band around the Γ point for $\mathbf{T}_A\mathbf{T}_C$ stacking aligns with previous results displayed in Fig. R3f extracted from ref (*Mater. Res. Express* **10**, 046302 (2023)).

For the calculations of supercell band and unfolded band in Figs. 4(b) and 4(d) in our initial manuscript, we used an identical crystal structure. For the $\sqrt{13}\times\sqrt{13}\times 4$ supercell, the vectors of the crystal lattice and reciprocal lattice are given by:

$$\begin{cases} \mathbf{a}' = 4\mathbf{a} + \mathbf{b} \\ \mathbf{b}' = -\mathbf{a} + 3\mathbf{b} \\ \mathbf{c}' = 4\mathbf{c} \end{cases}$$

and

$$\begin{cases} \mathbf{k}'_a = \frac{3}{13}\mathbf{k}_a + \frac{1}{13}\mathbf{k}_b \\ \mathbf{k}'_b = -\frac{1}{13}\mathbf{k}_a + \frac{4}{13}\mathbf{k}_b \\ \mathbf{k}'_c = \frac{1}{4}\mathbf{k}_c \end{cases}$$

We unfold the band structures from the $\sqrt{13}\times\sqrt{13}\times 4$ supercell to $1\times 1\times 1$ unit-cell. The unfolded spectral function is determined by $A(\mathbf{k}, E) = \sum_i |\langle \Psi_{i\mathbf{k}'} | \mathbf{k} \rangle|^2 \delta(\varepsilon_i - E)$, where $|\langle \Psi_{i\mathbf{k}'} | \mathbf{k} \rangle|^2$ represents spectral weight at \mathbf{k} , depending on whether ε_i corresponds to an eigenvalue at \mathbf{k} or not (*Phys. Rev. Lett.* **104**, 236403 (2010)). Applying the unfolding relation $\mathbf{k} = \mathbf{k}' + \mathbf{g}'$, where \mathbf{g}' is a reciprocal lattice vector of the $\sqrt{13}\times\sqrt{13}\times 4$ supercell, we can obtain the spectral function $A(\mathbf{k}, E)$.

In our initial manuscript, for better comparison with ARPES experiment results, we slightly adjusted the weight of the unfolded band to enhance the dispersion around -0.2 eV, resulting in some low-weight energy dispersions almost invisible. Here, we present the unfolded band structure plotted using the original weight in Fig. R4b. The hole-like band around the Γ point that disperses to -0.1 eV along Γ - M still has a very low weight. This result is consistent with the previously reported unfolded band by Lee et al. (*Phys. Rev. Lett.* **122**, 106404 (2019)) (also attached as Fig. R4d).

Fig. R4. Unfolded band structures for the $T_A T_C$ stacking, (a) plotted with adjusted weight for better comparison with ARPES experiment results shown in the manuscript, (b) plotted with original weight. (c) Unfolded band in the $k_z=\pi/4$ plane. (d) The unfolded band extracted from ref (*Phys. Rev. Lett.* **122**, 106404 (2019)), as ref 14 in the manuscript.

According to the above explanations regarding the folding relation between 1-layer and 4-layer supercells, the hole-like band of the 4-layer supercell will be unfolded from the $k_z^{4L}=0$ plane to the $k_z^{1L}=\pi/4$ plane of the 1-layer supercell. Then a further ab -plane unfolding from the $\sqrt{13}\times\sqrt{13}$ superstructure to 1×1 unit-cell occurs. To clearly illustrate this band, the calculated unfolded band along $M'-\Gamma'-M'$ in the $\pi/4$ plane is displayed in Fig. R4c. One can find that the hole-like band with a small unfolding weight appears in the $\pi/4$ plane.

Change to the manuscript:

- a) We update unfolded band structure in the Figure 4.

Figure 4. DFT calculated band structures.”

b) We inserted the following reference on page 8.

“The band structure for 4 layers 1T-TaS₂ (Fig. 4d) with David-stars T_AT_C stacking is in good agreement with previous reports.^{14,15,46}

⁴⁶ Zhang, W. & Wu, J. Stacking order and driving forces in the layered charge density wave phase of 1T-MX₂ (M = Nb, Ta and X = S, Se). *Mater. Res. Express* **10**, 046302 (2023).”

- 3) Even if the DFT calculations are correct, which seems to be the case at least for the unfolded band structures in Fig. 4 (d) and (h), the agreement with the ARPES data is, in my opinion, not as good as the authors claim. For instance the band at the Gamma point at -0.2eV in Fig 3 (e,f) is rather flat instead of electron like as the DFT calculation in Fig 4 (h) predicts. Perhaps some EDC could help to clarify this point more clearly. The electron-pocket like shape at Gamma is indeed a prominent characteristic of the TaTc stacking and therefore I believe it could well be that due to the sliding distortion a different stacking order occurs. Again, XRD could help to clarify this question.

Author reply:

We address both comment 3) and 4) collectively to avoid redundancy, as both center around concerns related to the consistence in the band dispersion between DFT calculations and ARPES in LC-TaS₂, which are interconnected with the David-stars stacking sequence, as highlighted in comment 1).

- 4) Also concerning the agreement between ARPES and DFT, I noticed that the k_z dispersion in the APRES data in Fig 3. (h) is nearly zero. The authors claim in line 204ff that this is in accordance with their DFT calculations. However, these calculations would predict a dispersion of roughly 0.1eV.

Author reply:

The less dispersive nature along both $k_{//}$ and k_z directions for $\mathbf{T}_A^S\mathbf{T}_C^S$ stacking compared to $\mathbf{T}_A\mathbf{T}_C$ stacking (cf. Fig. 4 in the manuscript, and also shown in this letter on page R18) from DFT calculations aligns with ARPES results, as correctly pointed out by the Reviewer.

However, we agree with the Reviewer that the calculated band dispersion for $\mathbf{T}_A^S\mathbf{T}_C^S$ stacking still appears larger than ARPES results for LC-TaS₂. This observation is further supported by EDCs shown below (Fig. R5).

Fig. R5. The energy distribution curves (EDCs) for 1T-TaS₂ and LC-TaS₂ extracted from Fig. 3b and 3e in the main text. The dashed lines and symbols represent the band positions.

Although XRD results (Fig. 1f, and also shown in this letter on page R7 and R22) clarify the absence of paired \mathbf{T}_A stacking, the existence of \mathbf{T}_C^S stacking is revealed by STM (cf. Supplementary Fig. S6, and also shown in this letter on page R11). However, the precise disordered CDW stacking of $\mathbf{T}_A^S\mathbf{T}_A^S$, $\mathbf{T}_A^S\mathbf{T}_C^S$, (as illustrated in Supplementary Fig. S9, and also shown in this letter on page R25) or any additional combinations cannot be definitively determined.

We thus left the disordered CDW stacking configuration in LC-TaS₂ as an open

question that warrants further investigation. Please refer to the **change to the manuscript** section addressing comment 1) for detailed revisions referred to disordered CDW stacking revealed by XRD.

Change to the manuscript:

We changed the following sentence on page 8 from

“In addition, the dispersion along the Γ - A direction is reduced to ~ 0.1 eV for $T_A^S T_C^S$ stacking from ~ 0.18 eV for $T_A T_C$ stacking, in excellent agreement with ARPES results. For 20 T -TaS₂, the dominant peak at ~ -0.2 eV exhibits less photon energy dependence (Fig. 3h).”

To

“In addition, the dispersion along the Γ - A direction is reduced to ~ 0.10 eV for $T_A^S T_C^S$ stacking from ~ 0.18 eV for $T_A T_C$ stacking. **The reduced band dispersion aligns with observations from ARPES results.** For LC-TaS₂, the dominant peak at ~ -0.2 eV exhibits less photon energy dependence (Fig. 3h).”

- 5) In line 231ff, a connection is made between the charge excitation gap and the transport properties at low temperatures. While this is correct in principle, it should be noted that the gaps derived from the Arrhenius diagram of electrical resistance are of the order of a few meV and thus much smaller than the gaps calculated in GGA+U or seen in STS. In general, the traditional GGA+U method may not be a good approach to describe the electronic structure of this compound, as it may be better described as a cluster Mott system, where the Mott-Hubbard U acts on the whole SOD cluster rather than on the single Ta site. Such calculations have been reported [here](https://iopscience.iop.org/article/10.1088/2053-1583/ace374/meta), for example:

Author reply:

We appreciate the Reviewer to bring this point to our attention. In this system, GGA+GOU may be a suitable approach to describe the electronic structure. Figure R6 represents the band structures calculated using the DFT+GOU method. The self-consistent \bar{U} was obtained through the ACBN0 method (*Phys. Rev. X* **5**, 011006 (2015)) implemented in the AFLOW π package (*Comput. Mater. Sci.* **136**, 76-84 (2017)); available from <http://afwlib.org/src/afwlp/>). Although the calculated gaps are comparable for DFT+GOU and GGA+U (65 meV and 52 meV for $T_A T_C$, and ~ 20 meV and 25 meV for $T_A^S T_C^S$ stacking), the obtained self-consistent \bar{U} of $\sim 0.85/0.86$ V for Ta-atoms we obtained is more than two times larger than the reported values of 0.33/0.45 eV. (*2D Mater.* **10**, 045001 (2023); *Phys. Rev. Lett.* **126**, 196406 (2021)). Some tips may have been overlooked, leading to this deviation. To ensure clarity and avoid any potential misleading, we have chosen to present our GGA+U results, which we are more familiar with. We may consider reporting the results obtained using the GGA+GOU method in further work once the associated issues are resolved.

Fig. R6. Band structures for (a) $T_A T_C$ stacking and (b) $T_A^S T_C^S$ stacking calculated by the DFT+GOU method. The self-consistently calculated \bar{U} parameters for Ta- and S-atoms are shown at the top of each panel.

A limitation of GGA+U method lies in the accuracy of gap value in this system, given its sensitivity to the on-site U value. To evidence the persistence of energy gap, calculations are performed with various U values. As shown in Fig. R7, a small gap persists even with the smallest $U=0.94$ eV, confirming the Mott nature of TaS₂ with a laddering stack structure.

Fig. R7. Band structures for the $T_A^S T_C^S$ stacking with (a) $U=0.94$ eV, (b) $U=1.94$ eV, and (c) $U=2.94$ eV, respectively.

Change to the manuscript:

We inserted the following sentences and references in the Experimental section on page 13.

“Notably, the GGA+U method has limitations regarding the accuracy of gap value in this system, given its sensitivity to the on-site U value. A potential alternative approach is the DFT+GOU method, where the self-consistent \bar{U} acts on the entire David-star cluster. This method may offer a more precise description of the electronic structure,^{19,42} warranting further theoretical investigation.

19. Shin, D. *et al.* Identification of the Mott insulating charge density wave state in 1T-TaS₂. *Phys. Rev. Lett.* **126**, 196406 (2021)

42. Dong, J. *et al.* Electronic dispersion, correlations and stacking in the photoexcited state of 1T-TaS₂. *2D Mater.* **10**, 045001 (2023)”

-
- 6) In the Discussion the authors conclude that the laddering stack structure weakens the interlayer coupling by reducing the wave function overlap in adjacent layers. However, in addition to the in-plane shift of ~ 0.17 Å, the XRD data indicate a reduction of the c -lattice parameter of about -0.03 Å. Was this shortening of the c -lattice parameter taken into account in the DFT calculations? I could imagine that this c -lattice parameter contraction, even if relatively small, may have a significant impact on the low-energy electronic structure. In this context, it is also important to note that even a relatively low hydrostatic pressure completely suppresses the CCDW phase and its characteristic TaTc stacking in favor of the NC-CDW phase in $1T$ -TaS₂, perhaps mainly due to the compression of the c -lattice parameter, since the lattice is much stiffer in the ab -plane than along c . Accordingly, the TaTc-CDW stacking of the CCDW phase is very sensitive to changes in the interlayer spacing.

Author reply:

We acknowledge the valid point that, assuming no in-plane shift occurs, the reduction of the c -lattice parameter indeed increases the interlayer coupling. (*Phys. Rev. B*, **90**, 045134 (2014)) The resulting loss of the David-stars stacking order explains the collapse of C-CDW phase into NC-CDW phase, (*Nat. Commun.* **11**, 4215 (2020)) in particular under low hydrostatic pressure. (*Nat. Mater.* **7**, 960-965 (2008); *Phys. Rev. B*, **87**, 125135 (2013)) The sensitivity of the T_AT_C stacking of the CDW phase to changes in the interlayer spacing is highlighted.

It is important to note that both c -lattice contraction and in-plane shift were taken into account in the DFT calculations. Supplementary Fig S8 (also attached below) displays the band structures for reduced c -lattice parameter, where c_0 represents the theoretically optimized lattice parameter of the T_AT_C stacking. Remarkably, despite these structural adjustments, the low-energy electronic structure exhibited negligible changes. GGA calculations reveal that the electron pocket at the L point experiences subtle enlargement with reduced c -lattice parameter, whereas the whole band dispersion remains almost unchanged. GGA+U calculation results demonstrate that as the c -lattice parameter decreases, the energy gap decreases slightly (Figs. S8g and S8h).

Our observations suggest that the laddering stack structure predominantly weakens the interlayer coupling. In this regards, the in-plane shift provide an alternative interpretation for the evolution of electronic structures beside out-of-plane contraction/expansion, considering the stiffness in the ab -plane.

Change to the manuscript:

- a) We inserted the following sentence on page 8 in the main text.

“For comparison purposes, the GGA+U method is also employed to calculate the band structure, as shown in Fig. 4e. As expected, the limited increase in the energy gap after the introduction of the Hubbard U term indicates that on-site Coulomb repulsion is not the primary factor affecting the energy gap for T_AT_C stacking. In addition, interlayer distance contraction has negligible influence on the low-energy

electronic structure beyond the reduced gap (Supplementary Fig. S8), confirming the dominant roles that ladder-like stack played in determining the interlayer coupling. The validity of simplifying the system to only 4 layers and employing a larger translational shift is discussed in the Methods part.

b) We displayed the band structures for reduced c -lattice parameters in Supporting Information.

Fig. S8. DFT calculated band structures with reduced c -lattice parameter. Band structures for $T_A T_C$ stacking with $\delta a = 0.53 \text{ \AA}$ and reduced c -lattice parameter, where c_0 represents the theoretically optimized lattice parameter of the $T_A T_C$ stacking. Remarkably, despite these structural adjustments, the low-energy electronic structure exhibited negligible changes. (a-c) GGA calculations reveal that the electron pocket at the L point experiences subtle enlargement with reduced c -lattice parameter, whereas the whole band dispersion remains almost unchanged. (d-f) GGA+U calculation results demonstrate that as the c -lattice parameter decreases, the energy gap decreases slightly. (g)(h) The gap value determined by the difference of the VBM and CBM. The negative gap indicates a metal state.”

7) The authors should definitely provide more experimental details about the annealing procedure to produce the 20T-TaS₂ structure to ensure reproducibility.

Author reply:

Similar concerns are raised by Reviewer #1 [comment 1)] and #3 [comment 3) and 4)], please refer to the **Author reply/Change to the manuscript** to comment 1) of Reviewer #1 on page R2 and the **Author reply** to comment 3) and 4) of Review #3 on page R39-42.

To summarize, I think the discovery of the 20T-TaS₂ structure is very interesting. If the authors were to describe the annealing protocol in more detail so that their results could be reproduced by others, I think it would certainly be worth publishing. However, the conclusion they have drawn regarding the persistence of TaTc stacking in 20T-TaS₂ and the strengthening of electron-electron correlations is not very solid in my opinion, and I cannot recommend publishing the manuscript in its current form.

Author reply:

We appreciate the possibility to publish this manuscript, addressing three crucial aspects: (i) the reproducibility of ladder structure; (ii) the persistence of $\mathbf{T}_A^s \mathbf{T}_C^s$ stacking in LC-TaS₂ experimentally and (ii) strengthening of electron-electron correlations the theoretically. We will elaborate on each point individually:

[redacted]

(ii) Regarding the precise disordered David-stars stacking configuration in LC-TaS₂ crystals, it remains an open question [see detailed to **change to the manuscript** section to comment 1)].

Although the order CDW \mathbf{T}_A stacking is ruled out by XRD, the \mathbf{T}_A^s and \mathbf{T}_C^s stacking configuration is observed by STM (Supplementary Fig. S6 on page R11 in this letter). The $\mathbf{T}_A^s \mathbf{T}_C^s$ stacking, adopted for electronic structures calculation of LC-TaS₂ crystals, aims to emphasize the role that atomic layers sliding play in altering the electronic structures. The precise disordered David-stars stacking configuration in LC-TaS₂, in particular persistence of $\mathbf{T}_A^s \mathbf{T}_C^s$ stacking in the bulk, cannot be addressed in the current manuscript. That is why we did not initially include \mathbf{T}_C^s assumption in Fig. 1g.

(iii) Addition efforts regard to DFT calculations with various CDW disordered stacking are dedicated to strengthening the electron-electron correlations. We hope our current efforts meet the satisfactory of the Reviewer.

[redacted]

Reviewer #3 (Remarks to the Author):

Yihao Wang et al. report the different structural and electronic properties of the 1T-TaS₂ and 20T-TaS₂. The latter has an unconventional stacking order, in which the David's stars are slightly displaced with respect to the 1T structure. An impressive amount different experimental methods are employed in order to characterize the samples as: STEM, XRD, STM, ARPES, Transport. By these means the authors convincingly show that the 1T-TaS₂ phase corresponds to the a dimerized band insulator whereas the 20T-TaS₂ is an undimerized Mott-insulator. The outcome is important and resolves controversial questions on a compound which is currently attracting strong interest. I am confident that the reported results will impact a broad community. The article is clearly written and most of it can be easily understood by non-specialists. Therefore, I recommend publication in Nature Communication.

Author reply:

We appreciate the positive feedback by Reviewer on our manuscript. Our goal was to employ a combination of microscopic (STEM, XRD, STM and ARPES) and macroscopic (transport) approaches to achieve a thorough characterization of the samples. We are pleased that the Reviewer found our approach convincing in establishing the dimerized band insulator nature of ideal 1T-TaS₂ and identifying LC-TaS₂ as an undimerized Mott-insulator.

The confidence in the impact of our reported results on the broader scientific community is truly motivating. We are grateful for the acknowledgment of the clarity of our writing and our efforts to make the content accessible to non-specialists. We sincerely appreciate the Reviewer's recommendation for publication in Nature Communications.

In the following I report some remarks that the authors should consider before publication:

- 1) The authors say "This value closely matches the theoretically predicted Mott-gap of ~ 0.2 eV for monolayer 1T-TaS₂". In reality, the STM of single layer 1T-TaS₂ has been already measured, and show a much larger gap than 0.2 eV (H. Lin et al., Nano Research 13, 133 (2020)). It is no surprise that there is no agreement between 20T-TaS₂ and the monolayer, mainly because of the screening of coulomb repulsion in a 3D structure. The authors should instead consider to cite the article: J. Dong et al., 2D Mater. 10, 045001 (2023), where ab-initio calculations with self-consistent screened U are calculated both for a 3D dimerized and 3D undimerized structure. The DOS of these calculations is in good agreement with the STM data measured by the authors for the dimerized 1T-TaS₂ and undimerized 20T-TaS₂.

Author reply:

Thank you for bringing relevant experimental (*Nano Res.* **13**, 33 (2020)) and theoretical (*2D Mater.* **10**, 045001 (2023)) sources to our attention, which were

inadvertently overlooked in the current manuscript. In fact, the article by *J. Dong et al.*, (*2D Mater.* **10**, 045001 (2023)) came to our attention during the manuscript submission process. We have now cited the mentioned articles and make the necessary adjustments accordingly.

Change to the manuscript:

We inserted the following sentence and reference on page 6.

“However, for the LC-TaS₂ crystal, two in-gap states appear at -0.06 eV and 0.18 eV, yielding a narrower energy gap of ~ 0.24 eV. These pronounced differences in spectrum features between 1T-TaS₂ and LC-TaS₂ crystals persist consistently across the surface with the on-top Davis-stars stacking configuration, as illustrated by line-cut maps in Supplementary Fig. S6. This values is closely matches the theoretically predicted Mott-gap of ~ 0.2 eV for monolayer 1T-TaS₂,¹⁰ indicating that this insulating state is induced by Mott localization. Nevertheless, this value is smaller than experimentally observed energy gap of ~0.45 eV in molecular beam epitaxy growth monolayer 1T-TaS₂ on a graphene/SiC surface,⁴¹ implying a larger screening of Coulomb repulsion in a 3D structure compared to a 2D monolayer. The STS spectra with a narrow energy gap have also been theoretically stimulated and experimentally observed on the cleaved 1T-TaS₂ surface with specific David-stars T_C stacking,^{13,20,21,42} suggesting their common origin of interlayer decoupling.

⁴¹ Lin, H. *et al.* Scanning tunneling spectroscopic study of monolayer 1T-TaS₂ and 1T-TaSe₂. *Nano Res.* 13, 133-137 (2020).

⁴² Dong, J. *et al.* Electronic dispersion, correlations and stacking in the photoexcited state of 1T-TaS₂. *2D Mater.* 10, 045001 (2023).”

- 2) Not being an expert in transport measurement, I do not understand the meaning of a negative carrier density and mobility in Fig. 5a,b). Moreover, it is not explained how the authors extract these numbers. Via Hall effect? Please clarify this point.

Author reply:

The carrier density and mobility values are extracted *via* the Hall effect. In Supplementary Figs. S12d and 12e, also attached below for your convenience, we display the magnetic field dependence of Hall resistivity ρ_{xy} for temperatures ranging from 10 K to 300 K. The negative slopes and positive slopes indicate the electron and hole carrier type, respectively. The extracted Hall coefficient R_H as a function of temperature is summarized in Fig. S12a. Subsequently, the corresponding carrier concentrations n and mobility μ are calculated using the formulas $n = (eR_H)^{-1}$ and $\mu = R_H/\rho_{xy}$, respectively, and summarized in Fig. 5a,b.

Change to the manuscript:

We inserted the following sentence on page 9 and Hall results in Figs. S12d and S12e.

“To explore the distinct macroscopic properties associated with this kind duality of insulating states, anisotropic electrical transport measurements were conducted. Figs. 5a and 5b show the temperature dependence of the mobility μ and carrier concentration n , respectively. The values were calculated using the formulas $n = (\epsilon R_H)^{-1}$ and $\mu = R_H/\rho_{xy}$, where ρ_{xy} and R_H represent Hall resistivity and Hall coefficient, as shown in Supplementary Fig. S12. The negative and positive signs denote the electron and hole carrier type, respectively.

Fig. S12 Anisotropic electrical transport properties. (a) The Hall coefficient R_H for the 1T-TaS₂ and LC-TaS₂. Temperature dependence of (b) original ρ_{ab} and (c) ρ_c , for 1T-TaS₂ and LC-TaS₂ upon cooling and warming. Hall resistivity ρ_{xy} as a function of the applied magnetic field at various temperatures for (d) 1T-TaS₂ and (e) LC-TaS₂.”

-
- 3) It is surprising that stacking disorder does not scramble a sliding occurring over 20 unit cells in the $20T$ structure. Moreover, the rapid quench to obtain $1T$ -TaS₂ is generally done to avoid the transformation in $2H$ polytype. How does it come that unquenched TaS₂ acquires the $20T$ structure instead of the $2H$ one?

Author reply:

The continue sliding over 40-unit cells in LC-TaS₂ indeed surprised us as well. While such continue sliding is observed in the T -phase, it is noteworthy that stacking disorder/fault interfere with the sliding in the trigonal prismatic H -phase (refer to Fig. R8 on page R36 of this letter), making a challenge in determining the periodicity on unis cell number.

We apologize for any confusion in the description of the synthesis of LC-TaS₂. Experimentally, we initially employ rapid quenching to obtain $1T$ -TaS₂ crystals and avoid transformation into the $2H$ polytype, a common practice. The obtained $1T$ -TaS₂ crystals are subsequently annealed gently, enabling the relaxation of quenched high-temperature $1T$ -phase towards the low temperature stable $2H$ -phase. The observed laddering stack structure is one of the intermediate phases in this transition. While its role in the $1T$ -to- $2H$ structural phase transition is beyond the scope of current paper, we plan to address it comprehensively in a parallel paper.

Change to the manuscript:

For a detailed synthesis procedure of LC-TaS₂, which is also concerned by Reviewer #1 [comment 1)] and #2 [comment 7)], please refer to the **Author reply** and **Change to the manuscript** sections to comment 1) of Reviewer #1 on page R2.

- 4) The authors do not talk about repeatability of the experiment. Are the samples from the same batch always showing the similar properties? Is the STEM looking the same after every cut? Does the STM depend on the sample position (this is indeed often reported!)?

Author reply:

We understand the Reviewer's concern regarding the repeatability of experiment on LC-TaS₂ crystals, and similar concerns are raised by Reviewer 1# [comments 1), 2) and 5)]. To avoid redundancy, please refer to **Author reply** and **change to the manuscript** sections to those specific comments (on page R2-5 and R9-11) for a detailed response, along with corresponding changes in the revised manuscript,.

The Reviewer is right in noting that both a larger band gap and smaller band gap coexist can coexist on $1T$ -TaS₂ surface. (*Phys. Rev. Lett.* **129**, 016402 (2022); *Nat. Commun.* **11**, 2477 (2020); *Phys. Rev. B* **105**, 035109 (2022)) Supplementary Figs. S6 c-f presents a series of STS curves collected at various locations on the surface of two LC- and $1T$ -TaS₂ crystals. Although there are slight differences in the spectra profiles for LC-TaS₂ (Fig. S6c), it is observed that the edge-to-edge energy gap shows less dependence on the sample position for the LC-TaS₂ crystal (Fig.

S6d with logarithmic scale). Additionally, the large band gaps can be consistently reproduced on our 1T-TaS₂ surface (Fig. S6f).

These results confirm the uniformity in the edge-to-edge gap for LC-TaS₂, which is attributed to the undimerized Mott-insulator nature of the ladder-stacked structure.

Change to the manuscript:

- a) We update the Fig. 1c to logarithmic scale to emphasize the edge-to-edge energy gap which is reproducible and inserted the following sentence on page 6 accordingly.

“Interestingly, the profiles of the corresponding dI/dV spectra change dramatically (Fig. 2c). For the as-grown 1T-TaS₂ crystal, two distinct peaks centered at approximately -0.19 eV and 0.26 eV feature a peak-to-peak energy gap with $\Delta \sim 0.45$ eV and an edge-to-edge gap of ~ 0.22 eV, consistent with previous experimental results.^{8,9,11,21,39,40} However, for the LC-TaS₂ crystal, two in-gap states appear at -0.06 eV and 0.18 eV, yielding a narrower energy gap of $\Delta \sim 0.24$ eV and an edge-to-edge gap of ~ 0.08 eV. These pronounced differences in edge-to-edge gap feature between 1T-TaS₂ and LC-TaS₂ crystals persist consistently across the surface, as shown in Supplementary Figs. S6d and S6f.

Figure 2. Evolution of chemical composition and electronic structures. c Spatially averaged dI/dV spectra with logarithmic intensity scale conducted at David-star centers for 1T-TaS₂ and LC-TaS₂ crystals at 4.5 K.”

- b) We displayed the STS curves at various locations on LC-TaS₂ and 1T-TaS₂ surface in the supporting information.

“

Fig. S6 A series of dI/dV spectra with (c)(e) linear intensity scale and (d)(f) logarithmic intensity scale measured at different surface locations of LC-TaS₂ and 1T-TaS₂.”

- 5) In the temperature dependent ARPES data, I expect the 1T-TaS₂ to show a transition to the 20T-TaS₂ upon heating and just before entering the NC CDW phase (see ref. 22 of your manuscript and J. Dong et al., *Physical Rev. B* 108, 155145 (2023)). Is this happening?

Author reply:

The references (ref. 22 and J. Dong et al., *Phys. Rev. B* 108, 155145 (2023)) imply that a transition from the band-insulator to intermediate Mott-insulator phase occurs upon heating from C-CDW to NC-CDW phase within a narrow temperature window (217-230K). This transition does not occur upon cooling from NC-CDW to C-CDW phase. These imply that the observed Mott-insulating state upon heating is an in-equilibrium state due to the competition between interlayer dimerization with David-stars stacking order, interlayer electron-electron correlations without David-stars stacking order and $e-ph$ coupling, (*Phys. Rev. B*, 107, 195401 (2023)) most likely resulting from non-thermal-equilibrium induced surface structural distortion associated with out-of-plane contraction in the low temperature CDW phase and the resulting out-of-plane CDW displacement. This explains the absence

of a Mott-insulating state upon cooling from high temperature NC-CDW phase.

To avoid this transition, we conducted the temperature dependent ARPES data by cooling the samples from 280 K to 20 K. The flat band, referred to as the Mott-insulating state, is not observed in $1T$ -TaS₂ surface, which is in-line with ref (*J. Dong et al., Phys. Rev. B*, **108**, 155145 (2023)). In contrast, the Mott-insulator associated flat band is observed at the NC-CDW to C-CDW transition temperature for LC-TaS₂ and persists at lower temperatures, indicating the intrinsic Mott-insulator nature of LC-TaS₂. That is, once the periodic 40-layer laddering structure is obtained, a temperature independent Mott-insulator would be expected. However, this is not the case from the ref (*Phys. Rev. B*, **108**, 155145 (2023)). Therefore, we do not expect the $1T$ -TaS₂ to show a transition to the LC-TaS₂ upon heating just before entering the NC-CDW phase.

Additional two experimental evidences are shown below:

(i) The interlayer distance for the intermediated Mott-insulator phase occurred at 217-230K (ref 22) is similar to that of NC-CDW phase at room temperature as revealed by constant (004) diffraction peak ($\sim 63^\circ$) upon heating shown in Fig. 5a and 5b of ref 22 and attached below as Fig. R9. This is against our results that the interlayer distance for the ladder stack structure is 0.03 Å smaller than that for $1T$ -TaS₂ determined from synchrotron based XRD (Fig. 1c in our manuscript and attached below).

Fig. 1c Synchrotron-based XRD spectra around the (001) diffraction peak displayed in the manuscript.

Fig. R9 (a) XRD intensity of 1T-TaS₂ measured along *c*-axis at 120 and 300 K. Inset panels show the emergence of (0, 0, 5/2) and (0, 0, 7/2) peaks using 1100 times expanded scales. (b) Temperature-dependence of the (0, 0, 7/2) and (0, 0, 4) peaks upon heating. (c) Schematic illustration of the interlayer dimerization.

Represented from ref (*Nat. Commun.*, **11**, 4215 (2020)).

(ii) CDW stiffness is reported for surface intermediate Mott-insulator phase compared to bulk (*J. Dong et al., Phys. Rev. B* **108**, 155145 (2023)). The CDW amplitude mode for surface intermediated insulator is ~ 60 GHz stiffer than for bulk in the C-CDW state (140K, see Fig. 4d and 4e in ref *Phys. Rev. B* **108**, 155145 (2023) and represented below as Fig. R10). This observation is in clear contrast to our Raman results shown in Fig. R11. All in-plane E_g modes associated vibrational peaks of CDW remain unchanged, whereas all out-of-plane A_g modes associated vibrational peaks (excepted one at ~ 229 cm⁻¹) undergo red shift in LC-TaS₂ crystals, which indicate out-of-plane CDW softness (Fig. R11). (Further analysis is needed to provide robust evidence)

Fig. R10. (d) Fourier transform of the CDW amplitude mode measured by time-resolved photoemission (*surface sensitive*) at 140 K. (e) Fourier transform of the CDW amplitude mode measured by transient reflectivity (*bulk information*) at 140 K.

Represented from ref (*Phys. Rev. B* **108**, 155145 (2023)).

Fig. R11. Raman spectra for 1T- and LC-TaS₂ crystals at C-CDW states of 50 K.

Although the periodic 40-layer laddering structure, which is obtained at higher temperature of 150-200 °C, is not expected upon heating from C-CDW to NC-CDW phase in 1T-TaS₂ at roughly -40°C, the potential for interlayer atomic sliding confined to the surface region (5-8 layers) cannot be ruled out. Recently, irregular interlayer atomic sliding in layered bulk PbI₂ at 120 K have elucidated changes in photoconductivity and photoluminescence, as represented in Fig. R12. (*Nat. Commun.*, **14**, 1981 (2023))

However, it is of great challenge to provide direct evidence due to:

- (i) STM cannot resolve the tiny out-of-plane misalignment (0.015 nm) in the

C-CDW states due to the loss of atomic resolution;

(ii) During the cutting of STEM samples, the deposited Pt layer on surface of sample may introduce artificial factors;

(iii) In-situ STEM measurements upon heating are limited in representing the thermal gradient along the c -direction for ARPES measurement, given the distinct dimensionality of samples. That is, extension along ab -plane for ARPES measurements and along c -direction for STEM measurements.

Change to the manuscript:

We inserted the following sentences and references on page 7 in the main text.

“These characteristics were similar to those reported in $1T$ - TaS_2 during a band-to-Mott insulating phase transition triggered by temperature.²² In contrast, the observed characteristics in LC- TaS_2 (Fig. 3f and Supplementary Fig. S3d) remain unchanged at 20 K, indicating the insensitivity of the electronic states to temperatures and confirming their intrinsic nature. This feature, along with distinct interlayer distance difference between LC- and $1T$ - TaS_2 crystals, implies that the periodic laddering stack structure is unlikely the structural origin of the

heating-triggered surface intermediate Mott-insulator phase.^{22, 44} However, the possibility of an interlayer atomic sliding confined to the surface region (5-8 layers), such as the irregular sliding with observed in layered bulk PbI_2 ,⁴⁵ cannot be ruled out.

²² ang, Y. D. *et al.* Band insulator to Mott insulator transition in 1T-TaS₂. *Nat. Commun.* **11**, 4215 (2020).

⁴⁴ Dong, J. *et al.* Dynamics of electronic states in the insulating intermediate surface phase of 1T-TaS₂. *Phys. Rev. B* **108**, 155145 (2023).

⁴⁵ Cha, S. *et al.* Order-disorder phase transition driven by interlayer sliding in lead iodides. *Nat. Commun.* **14**, 1981 (2023).”

REVIEWERS' COMMENTS

Reviewer #1 (Remarks to the Author):

I have carefully read the Authors' extensive responses to my comments and to those of the other Referees. Overall, I am very satisfied with the Authors' responses to my queries and criticisms, and with the revisions to the paper. I especially appreciate that the Authors have included a detailed description of how other groups can attempt to reproduce the LC-TaS₂ samples.

I also understand and share the feeling of the second Referee, that it would have been satisfying to see more evidence for the bilayer stacking pattern of the 3d CDW in the LC-TaS₂ samples. Nevertheless, the Authors openly acknowledge that this is an open question, and given the abundance of interesting data and observations in this paper, I am still inclined towards recommending the paper for acceptance.

I congratulate the Authors on their very interesting findings.

Reviewer #2 (Remarks to the Author):

I studied the extensive rebuttal letter and the revised manuscript with great interest. The authors have indeed thoroughly addressed all points raised in my previous report. I appreciate the extensive additional calculations, explanations and experimental data which, in my opinion, significantly improve the robustness of the presented results.

However, I still have one request regarding the XRD measurements: I have made a very naive estimate for the XRD intensity of some super lattice reflections associated to the 40 times larger c-lattice parameter of the sliding structure. I assumed a super cell of 40 undistorted 1T-TaS₂ layers which are stacked with a slide vector $t = -1/40 * (a+2b) + c$ as suggested by the authors and calculated the structure factors. It turns out that there are some peaks with non-integer l component (in Miller indices corresponding to the regular single-layer structure) with significant intensity. For instance the peak (1 0 21/20) should have an intensity of ~25% of the (0 0 1) peak. Note that in the other in-plane direction the peak occurs at (0 1 1). As the XRD probes a rather large volume it is probably reasonable to assume that multiple sliding domains would occur. Accordingly one should experimentally observe not only the (1 0 1) but also a satellite at (1 0 21/20) with comparable

intensity. Depending on the experimental resolution, this satellite can of course be just a shoulder or a broadening of the peak. However, it is important to note that pure l-peaks such as (0 0 1) would not have satellites which provides an opportunity to experimentally search for systematic differences in peak width. Moreover, as the in-plane component of the XRD reflection increases, the distance between the satellite and the main reflection would increase and eventually two separate reflections should be observable. I think it is really worth looking for such signatures, as they would provide very direct confirmation of the sliding structure from bulk sensitive XRD.

Having said this, I think that even without this additional data the manuscript can be published in Nature Communications.

Reviewer #3 (Remarks to the Author):

The authors have convincingly answered all my concerns. I therefore recommend publication of the manuscript.

Point-by-point responses to Reviewers of manuscript NCOMMS-23-42393-A

Comments in black –Author reply in blue–Revisions in purple with yellow highlight.

Reviewer #1 (Remarks to the Author):

I have carefully read the Authors' extensive responses to my comments and to those of the other Referees. Overall, I am very satisfied with the Authors' responses to my queries and criticisms, and with the revisions to the paper. I especially appreciate that the Authors have included a detailed description of how other groups can attempt to reproduce the LC-TaS₂ samples.

I also understand and share the feeling of the second Referee, that it would have been satisfying to see more evidence for the bilayer stacking pattern of the 3d CDW in the LC-TaS₂ samples. Nevertheless, the Authors openly acknowledge that this is an open question, and given the abundance of interesting data and observations in this paper, I am still inclined towards recommending the paper for acceptance.

I congratulate the Authors on their very interesting findings.

Author reply:

We are deeply grateful for the thorough review and positive feedback provided by Reviewer on our responses and revisions to the manuscript. Your acknowledgment of the effort we have put into addressing your queries and criticisms is truly appreciated.

Your feedback on this interesting finding, and inclination towards recommending the paper for acceptance is truly encouraging, and we are grateful for your support.

We aim to attract more researchers to this field by introducing the concept of translational degree of freedom in layered crystals to unlock fascinating physics (*Phys. Chem. Chem. Phys.* **26**, 7988-8012 (2024)). Exploring the accurate stacking pattern of the 3d CDW could serve as an excellent starting point.

Change to the manuscript:

We inserted our prospective paper proposing stacking engineering in 3D crystal on page 10.

“This method opens up opportunities to investigate low-dimensional physics in layered 3D crystals,⁵³ expanding its applications beyond the realm of sliding ferroelectricity.”

⁵³ Wang, J., Cheng, F., Sun, Y., Xu, H. & Cao, L. Stacking engineering in layered homostructures: transitioning from 2D to 3D architectures. *Phys. Chem. Chem. Phys.* **26**, 7988-8012 (2024).

Reviewer #2 (Remarks to the Author):

I studied the extensive rebuttal letter and the revised manuscript with great interest. The authors have indeed thoroughly addressed all points raised in my previous report. I appreciate the extensive additional calculations, explanations and experimental data which, in my opinion, significantly improve the robustness of the presented results.

Author reply:

Thank you for Reviewer's diligent examination of our rebuttal letter and the revised manuscript. We are pleased to hear that we have successfully addressed all points raised in Reviewer's previous report.

However, I still have one request regarding the XRD measurements: I have made a very naive estimate for the XRD intensity of some superlattice reflections associated to the 40 times larger c -lattice parameter of the sliding structure. I assumed a super cell of 40 undistorted 1T-TaS₂ layers which are stacked with a slide vector $t = -1/40 * (\mathbf{a}+2\mathbf{b})+\mathbf{c}$ as suggested by the authors and calculated the structure factors. It turns out that there are some peaks with non-integer l component (in Miller indices corresponding to the regular single-layer structure) with significant intensity. For instance the peak (1 0 21/20) should have an intensity of ~25% of the (001) peak. Note that in the other in-plane direction the peak occurs at (011). As the XRD probes a rather large volume it is probably reasonable to assume that multiple sliding domains would occur. Accordingly one should experimentally observe not only the (101) but also a satellite at (1 0 21/20) with comparable intensity. Depending on the experimental resolution, this satellite can of course be just a shoulder or a broadening of the peak. However, it is important to note that pure l -peaks such as (001) would not have satellites which provide an opportunity to experimentally search for systematic differences in peak width. Moreover, as the in-plane component of the XRD reflection increases, the distance between the satellite and the main reflection would increase and eventually two separate reflections should be observable. I think it is really worth looking for such signatures, as they would provide very direct confirmation of the sliding structure from bulk sensitive XRD.

Author reply:

We deeply appreciate the Reviewer's detailed analysis and the proposed superlattice reflections of the sliding structure. We believe this issue can be resolved by synchrotron-based XRD, which offers higher resolutions and 2D detector for precise detection of the (101) peak.

During our investigation, we did observed the additional peaks in synchrotron-based XRD. However, we have chosen to withhold presenting these results at this stage due to the challenge in confirming their attribution, thus avoiding any potential for misleading interpretations.

To ensure the repeatability and origin of these peaks, we have applied

synchrotron beamtime for a detailed and systemic investigation. This process may take some time, but we are optimistic that the outcomes of this specific examination will warrant publication as a follow-up paper.

Having said this, I think that even without this additional data the manuscript can be published in Nature Communications.

Author reply:

We appreciate Reviewer's confidence in the publication potential of our manuscript even without the additional XRD data. Your support means a great deal to us, and we sincerely appreciate it.

Reviewer #3 (Remarks to the Author):

The authors have convincingly answered all my concerns. I therefore recommend publication of the manuscript.

Author reply:

Thank you for continued support by Reviewer. We are pleased that our responses have addressed all of your concerns. Your recommendation for the publication of our manuscript is great appreciated.